# Scales of transformations—Modelling settlement and land-use dynamics in late antique and early medieval Basel, Switzerland

**Michael Kempf**[1,2,3]*, **Margaux L. C. Depaermentier**[4]

1 Department of Geography, Physical Geography—Landscape Ecology and Geoinformation, University of Kiel, Kiel, Germany, 2 CRC1266—Scales of Transformation, Project A2 'Integrative Modelling of Socio-Environmental Dynamics', University of Kiel, Kiel, Germany, 3 Department of Archaeology and Museology, Masaryk University, Brno, Czech Republic, 4 Department of Provincial Roman Archaeology, Vindonissa Chair, University of Basel, Basel, Switzerland

* kempf@geographie.uni-kiel.de

**Data Availability Statement:** All data underlying the analysis are freely available from the internet and referenced in the paper. Overview maps: - Digital elevation data derived from the USGS

## Abstract

Multicomponent environmental models have increasingly found their way into archaeological research. Mostly, these models aim to understand human patterns as a result of past climatic and environmental conditions over long-term periods. However, major limitations are the low spatial and temporal resolution of the environmental data, and hence the rather static model output. Particular challenges are thus the number of chosen variables, the comprehensiveness of the explanatory parameters, and the integration of socio-cultural decision-making into the model. Here, we present a novel approach to generate annually resolved landcover variability using a broad variety of climatic, geological, hydrological, topographical, and dendrochronological data composites (Palmer Drought Severity Index (PDSI)). We analyze land-use and settlement capacity and vulnerability to estimate the socio-cultural transformation processes at Basel (Switzerland) during the Late Antiquity and the Early Middle Ages. Our results highlight the potential of the PDSI to predict local river run-off behavior from catchment analyses. The model enables to trace landcover as well as socio-cultural response to climatic variability and subsequent adaptation to trends in environmental vulnerability. This approach further helps to understand population dynamics in the periphery of the Roman administrative boundaries and to revise traditional archaeological narratives of large-scale population replacements during the so-called Migration Period.

## Introduction

The urban agglomeration of Basel played a key role in Late Antiquity and throughout the Early Middle Ages. Basel was an important *castrum*, directly located at the river Rhine *limes* [1–3], and had central military, economic, political, and religious functions from the pre-Roman Iron Age onwards [1–3]. But our understanding of the area is considerably limited due to the overall lack of direct evidence of settlement features in the archaeological record–which in turn relies on graves and burial grounds [1, 4]. The knowledge about settlement structures,

(GMTED2010, Global Multi-resolution Terrain Elevation Data 2010, DOI: /10.5066/F7J38R2N) - Hydrologic streamflow derived from Open Street Map (https://wiki.openstreetmap.org/wiki/Main_Page) - Administrative boundaries derived from the European Commission (EN: © EuroGeographics for the administrative boundaries, https://ec.europa.eu/eurostat/web/gisco/geodata/reference-data/administrative-units-statistical-units/countries) - Geological map of Germany derived from INSPIRE under Creative Commons Attribution 4.0 International (CC BY 4.0) license (https://download.bgr.de/bgr/Geologie/GK1000-INSPIRE/gml/GK1000-INSPIRE.zip) Analysis: - High resolution digital elevation model (DEM) was acquired from the Bundesamt für Landestopografie Schweiz (swisstopo, grid size 0.5 m, https://www.swisstopo.admin.ch/de/geodata/height/alti3d.html) - A detailed groundwater map of Basel was redrawn from WFS data (https://wms.geo.bs.ch/) - Old World Drought Atlas (http://drought.memphis.edu/OWDA/).

**Funding:** MK received funding from the European Commission and the Masaryk University, Brno; Czech Republic, grant number CZ.02.2.69/0.0/0./18_053/0016952; Postdoc2MUNIm order number 21 0053 MK received funding from the German Research Foundation (DFG) and the CRC1266 Scales of Transformation, Kiel University, grant number 290391021 MLCD received funding from the University of Basel and the Swiss National Science Foundation (SNSF), grant number 100011 208060 The funders had no role in study design, data collection and analysis, decision to publish, or preparation of the manuscript.

**Competing interests:** The authors have declared that no competing interests exist.

however, is a crucial point to understand the socio-economic organisation of past societies, chronological sequences, and how these patterns were embedded into the broader regional context. In fact, very little attention has been put on how these social groups interacted among each other and how their integration into the local and regional landscape could have been manifested. Major questions are, for example, how were local activity spheres of late antique societies constructed at Basel? And what environmental parameters could have controlled the formation of contemporaneous landscapes? Because late antique and early medieval burial grounds were located outside of but close to the settlement [5–7], they can be used as indicators for the rarely preserved settlements. Moreover, recent studies showed that the location of settlement and corresponding burial ground would usually be based on pragmatic choices driven by basic needs, such as saving arable lands for agricultural practices or avoiding flooded areas [8–11].

In this article, qualitative and quantitative environmental analyses of late antique and early medieval graveyard catchments at Basel aim to model the potential distribution of corresponding settlements, associated croplands and pastures. The model considers environmental variability at the very local level in an area that is strongly impacted by continuous human occupation and urban spread. This is making it particularly difficult to trace past human activity spheres and to put them into a meaningful context under consideration of past environmental conditions. The strong surface and subsurface changes, that came with drainage activity, channelization, and urbanisation of the lower parts of the river Rhine floodplain and the tributaries [12], can only provide a fuzzy basis for qualitative and quantitative environmental catchment analysis. However, site-specific samples with high stratigraphic and chronological resolution, such as coring profiles, offer the possibility to transfer local data to a wider area under the premises of environmental system functionalities. Within such a system, the location of the archaeological record, can be used to identify potential palaeo-environmental surface and sub-surface conditions and to interpret potential preferences of site location parameters in the catchment of a site, such as elevated areas, particular geological and pedological units, or groundwater levels [13–15].

For this reason, a variety of environmental parameters were integrated to identify suitable settlement spots based on geological and pedological attributes, groundwater level anomalies and aquifer thickness, as well as the potential premodern hydrologic system at Basel. Together with the first attempt to reconstruct river Rhine run-off behaviour at Basel, modelled from PDSI variability (Palmer Drought Severity Index [16]) and climate fluctuations in Central Europe [17–19], a comprehensive model has been worked out to trace land-use suitability during the period 100–800 AD at Basel. We assume that a rather small activity sphere of late antique farming groups best predicts a self-sufficient crop cultivation strategy built on the small population numbers per generation, derived from the size of the graveyards and the chronological occupation of the sites. In most cases, quantitative models prevail, simply due to the large extent of the research area and the rather cost-intensive techniques of qualitative sampling [20]. Here, the small study area and the individual site approach favour a GIS- and R-software-based qualitative evaluation of land suitability. R environment: The R Project for Statistical Computing, https://www.r-project.org/about.html, last accessed, 28th of June 2022; QGIS, https://www.qgis.org/en/site/, last accessed, 28th of June 2022.

## Material and methods

A broad variety of data has been used and generated to estimate potential land-use and settlement locations during the late antique and early medieval occupation at Basel. The following section provides an overview of the archaeological background and the chronological

development at Basel and introduces into the environmental data and system analysis. Eventually, a detailed description of the landscape vulnerability model is given. Due to the workflow of the analyses, preliminary results from the environmental reconstruction are frequently intermixing with methodical considerations and are not presented in the results section.

## Archaeological background

The Münsterhügel played a significant economic and military role as Roman *oppidum* between 25 BC and 75 AD [21], but lost its strategic importance until the river Rhine became the border (*limes*) of the Late Roman Western Empire around 260 AD [22–24]. At that time, a *castrum* was built on the Münsterhügel together with other *castra* along the northern border of the *provincia Maxima Sequanorum* [25]. This *castrum*–and the entire river Rhine *limes*–was fortified again in the late 4[th] century AD, a period where a so-called *munimentum* (military fortification) was built to strengthen Basel on the northern riverbank [2, 3]. The site remained a military facility at least until the administrative collapse of the Western Roman Empire in the late 5[th] century AD [2]. However, the rural settlement development in this area is still widely unknown. Traditionally, it was assumed that the Roman population largely abandoned the region and was replaced at least on the Northern riverbank by barbarian groups that arrived in several waves of migrations or invasions between the 3[rd] and the 6[th] century AD [1, 26, 27]. During the late 5[th] to the early 6[th] century AD, the region was included into the Frankish Kingdom, but administration and infrastructure still relied on Late Roman legacies [28–30]. Thus, the city remained integrated within a considerable cultural and socio-economic network [31, 32]. The prosperity of the city in the 6[th] and 7[th] century AD is suggested by mintage on the Münsterhügel, the emergence of several settlements alongside important long-distance routes, and the development of a strong Christian administration [1, 33]. This led to a new political division of the region during the 7[th] century AD, with the manifestation of Frankish power and the diocese of Basel on the southern bank, and the Alamannic dukedom and the diocese of Constance on the northern riverbank of the river Rhine [1, 33]. The city soon became so important that the bishop's seat was transferred from Augst to Basel around the 7[th] or 8[th] century AD [1, 34] and the first attested cathedral was built by bishop Haito in the early 9[th] century AD [3].

## Historical framework

This historical framework has considerably influenced our perception of the transition between Late Antiquity and the Middle Ages at Basel. Consequently, people occupying the territories on the northern riverbank between the late 3[rd] and 6[th] centuries AD to dwell in and lay out burial grounds, were considered *non-Romans* or simply *Barbarians* with completely different cultural attribution, burial customs, diet, and material and immaterial affordances–hence manifestations of a *Germanic* lifestyle [1–3, 11]. These seemingly terminological as well as methodological accuracies of cultural definition based on the Roman-Barbarian-dichotomy, developed during the 18[th] century AD, have met with considerable opposition, and eventually led to the clash of ethnic identities, particularly in the discipline of Early Medieval Archaeology [28, 35–37]. Despite the persistent urge to ethnic attribution or the identification of individual origins inherent in manifold research approaches, recent scientific results from multidisciplinary analyses carried out at Basel and across Europe have emphasized the strongly local interaction patterns of local people with long-standing integrated and scattered origins during Late Antiquity and Early Middle Ages [38–40]. A major hurdle to simply remove the raison d'être of such a distinction in Basel and Europe is the simple fact that, to a large number, Late

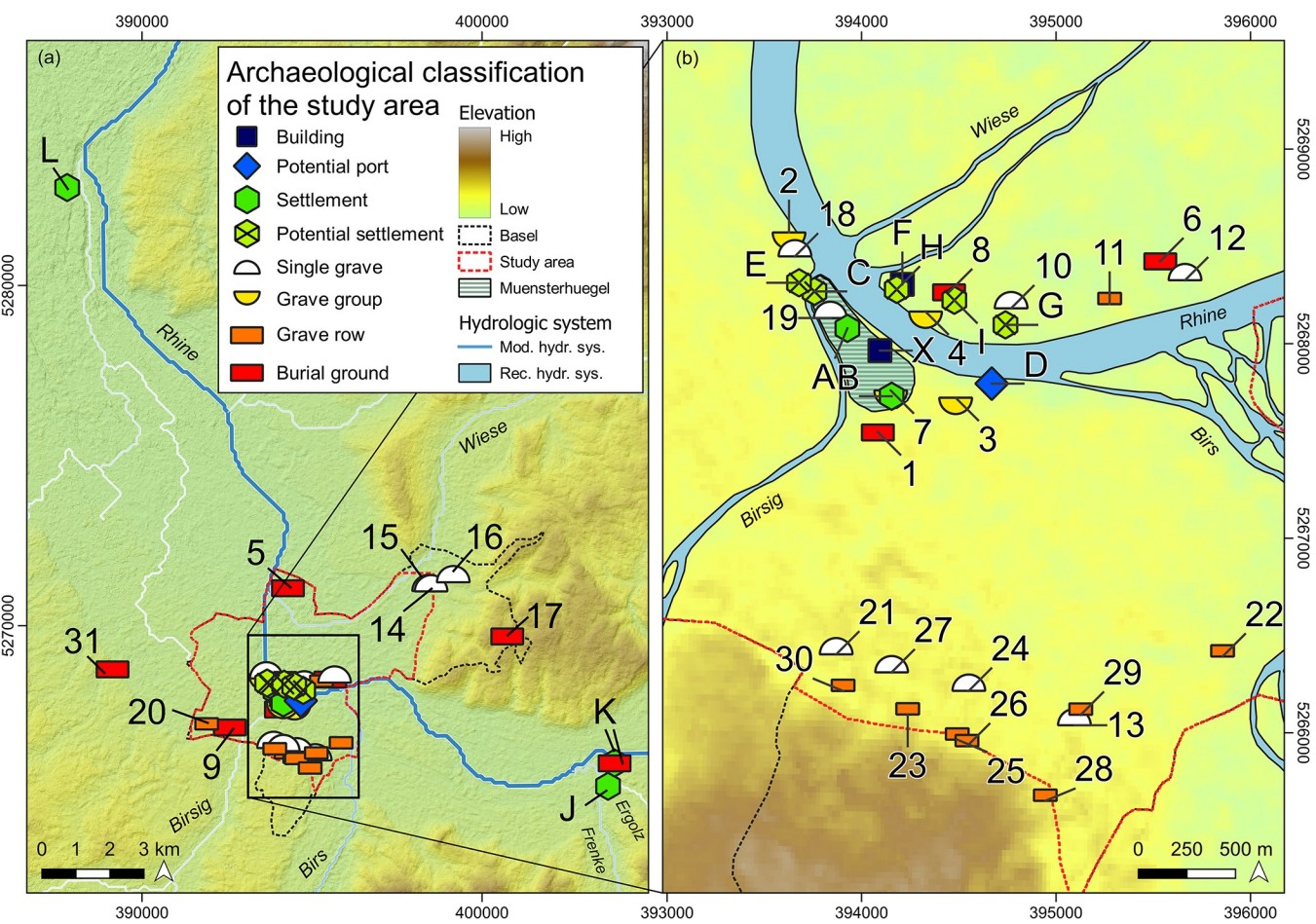

**Fig 1. Type of the archaeological record in the study area and the complementary region at Basel.** (see Table 1 for type, name, and chronological differentiation) a) complementary region coverage with topographic features and modern hydrologic system; b) zoom into the archaeological settings at Basel with topographic elements and reconstructed potential streamflow characteristics [75]. Images are for illustrative purpose only. Data source: hydrologic streamflow derived from Open Street Map (OSM, https://wiki.openstreetmap.org/wiki/Main_Page) data under https://creativecommons.org/licenses/by-sa/2. 0/, last accessed 3rd of November 2022). Administrative boundaries were individually redrawn using GIS software. Reconstructed streamflow was redrawn from Rentzel et al. (2015) (see ref. 75). Digital elevation data derived from the USGS (*GMTED2010*, Global Multi-resolution Terrain Elevation Data 2010 (GMTED2010) DOI number: /10.5066/F7J38R2N).

Antique and Early Medieval archaeological records consist of burial grounds, single graves, or grave groups and that corresponding settlement are largely missing [1, 4, 41] (see **Fig 1**).

## Settlement continuity during the transformation period

The transition between the Late Antiquity and the Early Middle Ages (i.e. between 250 and 400 AD) is traditionally associated with a demographic decline in the *regio Basiliensis* [42–44]. The implied decrease in land-use and settlement activities is suggested by the palynological and dendrological records [18, 45, 46], as well as by the small number of settlements known for this period [44]. However, the prevailing use of wood for building construction, the poor preservation of wooden structures in the archaeological context, and the possible continuity of settlement from the Early Middle Ages until today also influences the scarcity of settlement remains for this period [9, 11, 41, 47, 48]. Consequently, it is not surprising that there are only few traces of early medieval settlements in Basel. The most important–and the only well documented–settlement is located at the Münsterhügel [2]. The site was already occupied from at

least the protohistoric period onwards, but became particularly important during the Roman Imperial Period and again during Late Antiquity, with the first stone buildings and fortifications [3, 21, 25]. In Late Antiquity, it is assumed that the fortified Münsterhügel was rather loosely settled and that several areas were kept open to be used as gardens, small fields, or for animal husbandry [34, 49]. Even though both elite houses and buildings with a military or administrative function are known within the fortified area on the Münsterhügel during Late Antiquity and the Early Middle Ages, the evidence for the corresponding settlement structure remains scarce, most likely due to preservation issues [2]. Several early medieval pit houses are attested for the 7[th] and 8[th] centuries AD, and other finds and features indicate an extensive, loosely distributed, and rather hamlet-like settlement structure in this area, in which at least parts of the late Roman fortification were no longer standing upright [3, 50]. An extensive settlement outside the Münsterhügel is attested only from the 9[th] century AD onwards, for instance through the foundation of new churches [2].

A few late antique and early medieval artefacts have been found in thick, organic-rich, dark brown humus layers known as 'Dark Earth' in several parts of the present-day city, which are assumed to indicate a continuous settlement activity [49]. Besides the so-called *munimentum* on the northern riverbank, most late Roman traces are located in the area of the St. Alban-Vorstadt, where roads, buildings, and various other settlement structures could be identified, revealing also a possible continuous use of this area during the Early Middle Ages [51–53]. The area, situated at the confluence of the tributaries Birsig and the river Rhine on the southern riverbank further constitutes an important settlement spot [2]. Especially pottery sherds, coins, late Roman belt buckles, and crossbow brooches scattered at the Petersberg west of the Münsterhügel and on the left bank of the river Birsig were interpreted as evidence of a late Roman (military) settlement in this area [54]. The early medieval period provides even less direct evidence for settlements, consisting mostly of pottery fragments. Based on this, a potential settlement could be reconstructed in the area between the Burgweg and the Alemannengasse [55] as well as between the Mittlere Brücke and the Wettsteinbrücke [55, 56]–though these settlement records were mostly dated between the 7[th] and the 9[th] century AD. Moreover, an increase in agricultural activities in the region can also be deduced from the botanical remains of the middle of the 7[th] century AD onwards, which is traditionally explained by population growth [18, 45, 46]. Regarding the later stage of the early medieval period, settlement activity can only be attested indirectly through the presence of burial grounds. The most important limitation concerning the burials is that, even though the archaeological and typochronological dating is to a certain extent quite reliable [57], most of the burials at Basel (nearly 70%) do not contain any (chronologically relevant) grave goods and remain therefore undated. One should hence consider that the archaeological record is particularly biased by chronological uncertainty.

## Late antique and early medieval gravescapes

Late Antique burials dating to the late 3[rd] and the 4[th] century AD, are mostly known from the southern bank of the river Rhine (see **Fig 1** and **Table 1**). These include the burial grounds at Basel-Totentanz (n = 16) [54, 58], the early stages of the cemetery at Aeschenvorstadt ($n_{LATEANTIQUITY}$ = 49) [59] as well as a small burial group in the St. Alban-Vorstadt (n = 16) [60–62]. Few graves from the Petersgasse could also belong to this period [54, 58]. A recent study revealed that people were also buried between 370 and 415 AD at Basel-Waisenhaus (n = 11) [40, 56]. This small burial group is not only located just in front of the Münsterhügel on the northern riverbank, but also less than 200 m to the east of the *munimentum*. These are the only burials that can clearly be dated to the first half of the 5[th] century AD in Basel. The continuous burial activity in Basel-Aeschenvorstadt ($n_{TOTAL}$ = 585) during this period is mostly

**Table 1. Position, site id, label, name, type, and chronological occupation of the archaeological record discussed in this article (see Fig 1 for geographic location).**

| lon | lat | id | name | cat | NMI | R | LA | 5th | 500 | 6th | 600 | 7th | 7th/8th |
|---|---|---|---|---|---|---|---|---|---|---|---|---|---|
| 7.592 | 47.552 | 1 | Aeschenvorstadt | BG | 585 | x | x | x | x | x | x | x | |
| 7.585 | 47.561 | 2 | Totentanz | GG | 16 | x | x | | | | | | |
| 7.597 | 47.554 | 3 | St. Alban-Vorstadt | GG | 16 | x | x | | | | | | |
| 7.595 | 47.558 | 4 | Waisenhaus | GG | 11 | | x | x | | | | | |
| 7.594 | 47.584 | 5 | Kleinhüningen | BG | 305 | | | x | x | x | x | x | x |
| 7.611 | 47.56 | 6 | Gotterbarmweg | BG | 38 | | | x | x | | | | |
| 7.593 | 47.554 | 7 | Antikenmuseum | GG | 10 | | | | x | x | x | | |
| 7.596 | 47.559 | 8 | St. Theodor | BG | 23 | | | | | x | x | x | |
| 7.572 | 47.547 | 9 | Bernerring | BG | 45 | | | | | x | x | | |
| 7.601 | 47.558 | 10 | Burgweg | SG | 1 | | | | | | x | | |
| 7.607 | 47.558 | 11 | Grenzacherstrasse 127 | GR | 5 | | | | | | | x | |
| 7.613 | 47.559 | 12 | Grenzacherstrasse Solitude | SG | 1 | | | | | | | x | |
| 7.606 | 47.539 | 13 | Wolfgottesacker | SG | 1 | | | | | | | x | |
| 7.649 | 47.584 | 14 | Riehen Dorfkirche | SG | 1 | | | | | | | | x |
| 7.65 | 47.584 | 15 | Riehen Baselsstrasse 46 | SG | 1 | | | | | | | | x |
| 7.658 | 47.587 | 16 | Riehen Bosenhaldenweg | SG | 1 | | | | | | | | x |
| 7.68 | 47.573 | 17 | Bettingen | BG | 75 | | | | | | | | x |
| 7.586 | 47.56 | 18 | Petersgasse | SG | 1 | | | | | | | | x |
| 7.588 | 47.557 | 19 | Martinsgasse | SG | 1 | | | | | | | | x |
| 7.563 | 47.546 | 20 | Neuweilerstrasse | GR | 5 | | | | | | | | x |
| 7.589 | 47.542 | 21 | Pfeffingerstrasse | SG | 1 | | | | | | | | x |
| 7.616 | 47.542 | 22 | Steinenberg | GR | 5 | | | | | | | | x |
| 7.594 | 47.539 | 23 | Gundeldingerstrasse 315 | GR | 5 | | | | | | | | x |
| 7.598 | 47.54 | 24 | Laufenstrasse | SG | 1 | | | | | | | | x |
| 7.598 | 47.538 | 25 | Gundeldingerstrasse 394 | GR | 5 | | | | | | | | x |
| 7.598 | 47.538 | 26 | Gundeldingerstrasse 406 | GR | 5 | | | | | | | | x |
| 7.593 | 47.541 | 27 | Dornacherstrasse 192 | SG | 1 | | | | | | | | x |
| 7.604 | 47.535 | 28 | Reinacherstrasse | GR | 10 | | | | | | | | x |
| 7.606 | 47.539 | 29 | Gundeldingen | GR | 5 | | | | | | | | x |
| 7.59 | 47.54 | 30 | Gundeldingen | GR | 5 | | | | | | | | x |
| 7.526 | 47.562 | 31 | Hégenheim | BG | 46 | | | | | x | x | x | |
| 7.59 | 47.557 | A | Münsterhügel | SET | 0 | x | x | x | x | x | x | x | x |
| 7.592 | 47.556 | X | Cathedral | BLD | 0 | | | | | | | | x |
| 7.593 | 47.554 | B | St. Alban-Graben | SET | 0 | x | x | x | x | x | x | x | |
| 7.587 | 47.559 | C | Marktgasse | POT SET | 0 | x | x | | | | | | |
| 7.599 | 47.555 | D | Bottom Mühlenberg | POT PORT | 0 | x | x | | | | | | |
| 7.586 | 47.559 | E | Left bank Birsig/Petersberg | POT SET | 0 | x | x | | | | | | |
| 7.593 | 47.559 | F | *munimentum* | BLD | 0 | | x | x | | | | | |
| 7.6 | 47.557 | G | Burgweg/Alamannengasse | POT SET | 0 | | | | | | | | x |
| 7.593 | 47.559 | H | Mittlere-/Wettsteinbrücke | POT SET | 0 | | | | | | | | x |
| 7.597 | 47.558 | I | St. Theodor area | POT SET | 0 | | | | | | | | x |
| 7.72 | 47.533 | J | Augst | SET | 0 | x | x | | | | | | |
| 7.723 | 47.539 | K | Kaiseraugst | BG/SET | 1300 | x | x | x | x | x | x | x | |
| 7.504 | 47.689 | L | Kembs | SET | 0 | x | x | | | | | | |

assumed because the place was continuously used from the 1st to the late 7th century AD [59]. Some archaeological evidence suggests a possible early occupation of the sites Basel-Gotter-barmweg [63] and Basel-Kleinhüningen [64, 65], though the earliest burials with a reliable chronology date to the middle of the 5th century AD. Basel-Gotterbarmweg (n = 38) was already abandoned around 510 AD, while a new, only briefly occupied burial ground with particularly lavish grave goods was established around 540 AD and in considerable distance (ca. 1.7 km) on the southern bank of the river Rhine at Bernerring [66]. The burial ground Basel-Bernerring (n = 45) was only used until approximately 600 AD. The few graves from the courtyard of the Museum of Antiquities at Basel (Antikenmuseum, n = 10) also belong to the second half of the 6th century AD [67, 68]. Due to the rich foreign grave goods, archaeologists first assumed that high-ranking Germanic newcomers had deliberately buried their dead in distance to the burials at Basel-Aeschenvorstadt [3, 68].

Some burials scattered between the Theodorskirchplatz (n = 23) and the Wettsteinplatz (n = c. 5) on the northern bank of the river Rhine are also dated to the late 6th century AD [55, 69], which is only a few decades after the last individuals were buried in Basel-Gotterbarmweg. For this reason and despite the radical changes in burial practices at that time, it was long considered that these burials would correspond to the same community, who would have relocated its burial ground by approximately 1 km to the west [3]. With the recent archaeological discoveries, a link with the Basel-Waisenhaus burial group located less than 200 m to the south is also assumed–despite the chronological hiatus of more than a century.

At the end of the 6th and the beginning of the 7th century AD, a change in burial practices is documented in the whole region, including the Upper Rhine Valley and Southwestern Germany [70]. At Basel, this is visible through the abandonment of the large burial ground in Basel-Aeschenvorstadt, the emergence of separate burials in Basel-Kleinhüningen, and the spread of single burials all over the city, mostly along roads. Such graves are known from the Grenzacher-strasse, Neuweilerstrasse, Laufenstrasse, Münchensteinerstrasse, Gundeldingerstrasse, the Martinskirchplatz as well as from Riehen, and Bettingen [71]. The graves at St. Theodor were possibly already associated with an early and simple church [55], which constitutes a new and still unusual practice in the region for this period. Burials in and around the cathedral at the Münsterhügel did not occur until the late 7th or rather the 8th century AD, which would coincide with the transfer of the bishop seat from Augst to Basel during this period [1, 2, 34, 72]. Among the other burial places, a continuous use until at least the 8th century AD is only observed at the Theodorskirche and presumably also in Basel-Kleinhüningen ($n_{TOTAL}$ = 305).

The settlements have not been discovered yet, but it is usually presumed that they were located at strategic places. Basel-Totentanz and Basel-Aeschenvorstadt are located along the main road leading to the north, which is an antique long-distance road that connected Augst/ Kaiseraugst with Kembs [2]. Basel-Kleinhüningen and Basel-Gotterbarmweg could be linked to potential crossing points over the river Rhine [1, 32, 64]. Archaeological and topographical evidence also points to a possible crossing of the Rhine between St. Alban and the Burgweg during Late Antiquity [55] and a potential late Roman shipping port is considered to be located at the bottom of the Mühlenberg [62]. Basel-Bernerring is also strategically situated at an important road axis that was a connection across the Alps [66]. Several other burial grounds and settlements were founded in the 6th and 7th century AD along this long-distance road at the foothills of the Swiss Jura [1, 73], for example at Hégenheim (France) in the direct hinterland of Basel [74]. The scattered finds (such as weapons and jewellery) and later stone graves from the Gundeldingen quarter further demonstrate the important role of this road during the 7th century AD in Basel [3]. On the northern riverbank, the slab-covered graves from the 7th century AD at Grenzacherstrasse, close to Theodorskirche, and at Basel-Kleinhüningen seem to be located in accordance to an important long-distance road [55].

## Environmental settings

Modern Basel is located at the river Rhine, covering the northern and the southern parts of the high terraces of the river and the tributaries Wiese, Birs, and Birsig (Fig 2). The river Rhine is the largest drainage system in western Europe and drains geological formations of various origin and composition, such as crystalline rocks from the central alpine catchments, limestone

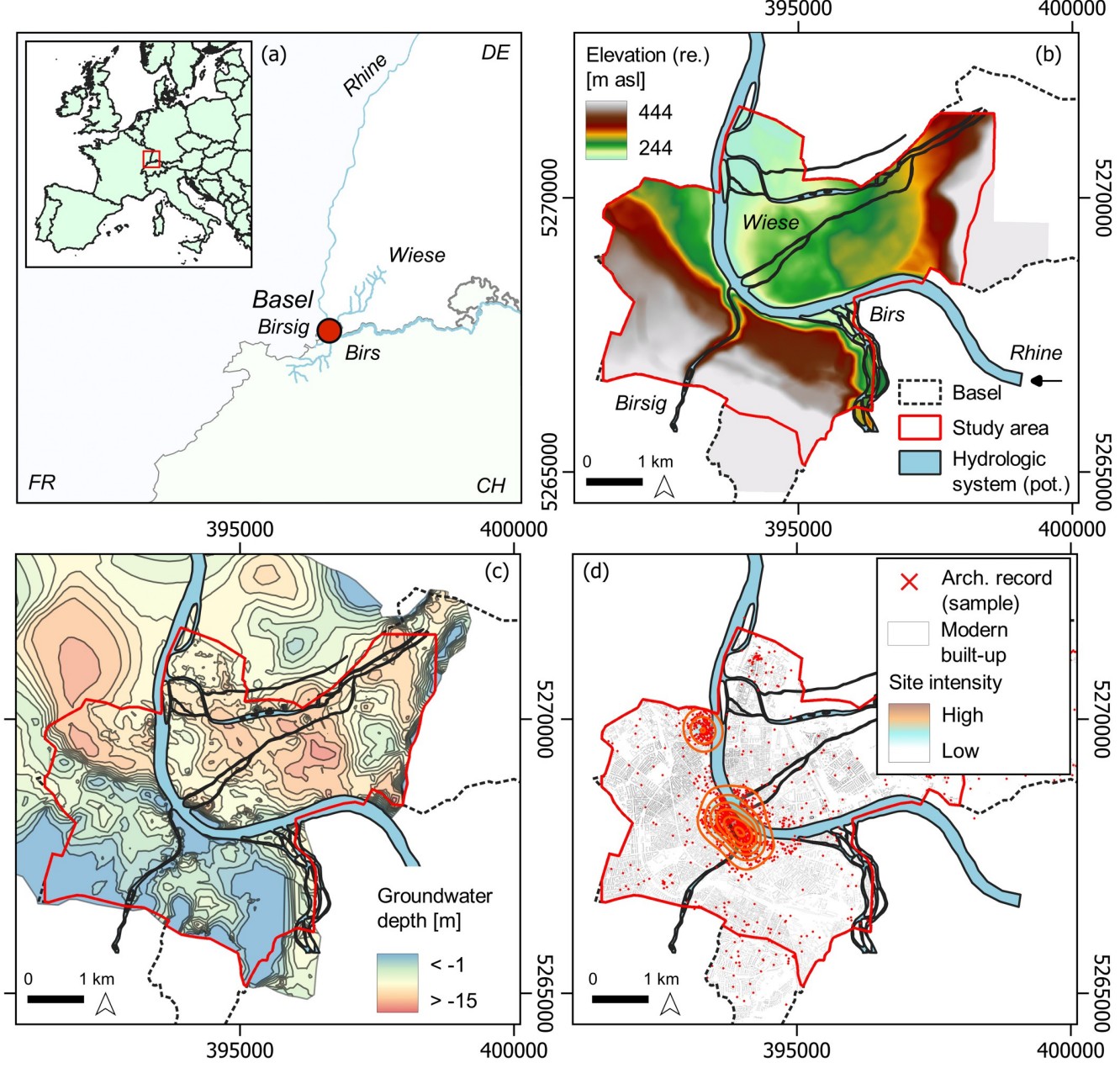

**Fig 2. Environmental and urban settings of the study area.** (a) Location of Basel (red circle); (b) reconstructed potential topographic and hydrologic conditions [75]; (c) groundwater level after drainage and channelization measures; (d) selected sample intensity of the total urban archaeological record at Basel (record excluding common era). Images are for illustrative purpose only. Data source: hydrologic streamflow derived from Open Street Map (OSM, https://wiki.openstreetmap.org/wiki/Main_Page) data under https://creativecommons.org/licenses/by-sa/2.0/, last accessed 3rd of November 2022). Administrative boundaries derived from the European Commission (EN: © EuroGeographics for the administrative boundaries, https://ec.europa.eu/eurostat/web/gisco/geodata/reference-data/administrative-units-statistical-units/countries, last accessed 21st of October 2022).

from Mesozoic cover, and Molasse from High Rhine valley [76, 77]. The fluvial characteristics are dominated by a nivo-glacial regime in the area of Basel, where the river enters the Cenozoic European Graben rift system [77]. The run-off maxima are controlled by early summer snow-melt discharge in the upstream regime, whereas the downstream regime is dominated by winter precipitation [78, 79]. In addition, the unregulated Lake Constance modulates the run-off maxima [80]. Particularly interesting is the location of Basel at the transition zone from the High Rhine fluvial regime towards the Upper Rhine Area (URA). The former is dominated by the river Rhine cutting into Mesozoic limestone formations of the Table Jura. In the URA, the earliest sedimentological record of the river system can be traced back to the Late Miocene, mostly dominated by sandy deposits [76]. According to Preusser (2008), the younger sediments of the southern graben are formed by coarse alpine meltwater deposits. At the junction at Basel, the river Wiese deposited silicate and limestone gravel (20% of the aquifer thickness) on top of river Rhine gravel (primarily limestone, 80% of the aquifer thickness) [81]. The sedimentation regime, however, has been constantly altered since the mid-Holocene, starting with Neolithic land-use and transformed sediment load from the tributaries [82]. At least since the late Iron Age and the Roman period in the region, a massive increase in colluvial deposits has been recorded, which was triggered by enhanced deforestation activity and agricultural exploitation further changing the sediment transportation regime of the entire system [79]. During the medieval period, dam constructions and flood protection measures were archaeologically recorded, which highlight the earliest river channelization strategies along the river Rhine. These peaked in the massive regulation activities during the 19th century, including the river Rhine tributaries at Basel [12].

Particularly the river Wiese, that drains the southern part of the Black Forest and enters the river Rhine at Basel is liable for heavy flooding events. The pronounced run-off gradient and the poor storage capacity of the geological formations in the catchment area, rapid flash floods occur during the winter months. Consequently, the run-off characteristics altered frequently during the Holocene, which resulted in constant relocations of the riverbed and the changes in the sedimentological regime [83]. Remnants of these riverbed relocations, such as medieval millponds, are still visible in the landscape and the urban layout of the neighboring cities Weil and Riehen [84]. The frequent flooding events and the shift to a gentle slope gradient after entering the extensive floodplain, further triggered a higher groundwater level, which rendered the area rather unsuitable for agricultural purposes. During the 14th century AD, partial channelization and drainage activity turned the humid marshes into meadows, pastures, and grasslands [85]. Eventually, the river experienced massive regulations and channelization activity during the past 19th and 20th centuries [83] to prevent flooding. Accordingly, the groundwater level dropped and the absence of flooding dynamics altered the floodplain composition [85].

## Datasets and processing

The following section includes a detailed description about the data preparation that enables the high-resolution environmental analysis and the model set-up at Basel.

**Topography.** A high resolution digital elevation model (DEM) was acquired from the Bundesamt für Landestopografie Schweiz (swisstopo, grid size 0.5 m, https://www.swisstopo.admin.ch/de/geodata/height/alti3d.html, last accessed 15th of April 2022) that covers the modern administrative boundaries of the Kanton Basel Stadt. The DEM comes pre-processed and smoothed with regards to infrastructure and built-up to prevent artefacts due to large artificial complexes of modern urban structures. However, broad areas of nowadays Basel are severely impacted by recent infrastructural mismanagement, which resulted for example in massively elevated motorway constructions, particularly affecting the northern part of Kleinbasel.

Furthermore, the tracks of the cargo railway are similarly elevated above the current surface level, giving the impression of broad linear structures in the DEM. Consequently, the data was harmonized, and large parts of the predominantly artificial surface changes were eliminated. Building on this, the DEM was further processed using focal statistics and the GRASS GIS module r.neighbors to smooth the surface by using a moving-window operation with average statistics and a circular neighborhood of 10 m [15, 86]. This operation was used to smooth small-scale data gaps after the built-up corrections considering the 0.5x0.5 m resolution of the DEM. Subsequently, the DEM was resampled to a 5 m grid size and the average smoothing algorithm was run again with a focal window of 100 m radius in a circular neighborhood. The DEM was then cropped to the maximum extent of the original data to delete edge effects of the focal statistics. To acknowledge for the potential landcover reconstruction and the reconstruction of the hydrologic system proposed by Rentzel *et al.* (2015), the riverbed of the river Birs was added to the DEM using a manually created streamflow DEM. Sandbanks within the river were adjusted gradually along the riverbed, rasterized, and smoothed using the above-mentioned modules from the toolbox (Fig 2B).

**Hydrologic system and groundwater variability.** A potential reconstruction of the pre-modern river Rhine run-off characteristics was taken from the visualization by van Holzen and partly prolonged towards the north-eastern part of the study area, following the topographical features of the DEM [75]. The reconstruction provides a potential model of the river Rhine tributaries prior to the massive channelization activity during the 19[th] century AD [12]. Particularly the streamflow characteristics of the river Wiese, draining the southern part of the Black Forest in modern Kleinbasel, represent an important feature in the palaeolandscape of the northern bank of the river Rhine. The river split into several branches on a broad and rather even floodplain before entering the river Rhine, which is considered to be located in its current riverbed. The uneroded location of prehistoric archaeological records strengthens the arguments of no larger channel reorganisation in the area at least since the Neolithic. Downstream of Basel, however, a pluvial regime created an anabranching and a widespread braided river system [12]. At Basel, the tributaries play a major role in riverbed formation, mostly due to continuous sedimentation from the northern Black Forest and the southern Jura Mountain range. Particularly the northern bank of the river Rhine is characterized by the river Wiese's run-off and sedimentation regime.

The aquifer thickness is particularly influencing the site suitability for settlement areas and agricultural crop production at Basel. To trace groundwater variability in the region, the detailed groundwater map of Basel was redrawn from WFS data (https://wms.geo.bs.ch/, last accessed 20[th] of April 2022). From the data, the strong local groundwater variability of the region becomes evident (see **Fig 2C**). Local anomalies push the aquifer upper limits close to the surface but drop significantly within short distance to up to 14–15 m below surface. Furthermore, the near-surface aquifer anomalies do not entirely correspond to the direction of the river Rhine run-off. Upstream of the reconstructed river deltas of the rivers Wiese and Birsig, a rather near-surface groundwater anomaly aligns with the river Rhine main channel. Downstream and at the confluence of the tributaries, a very high groundwater anomaly can be detected, which continuous in north-west direction–linked to palaeochannels cut into the pre-Quaternary bedrock [87, 88]. Due to riverbed developments in the vicinity of Kleinbasel, a more gentle transition between the riverbank and the terrace can be assumed for premodern periods, which favoured the accumulation of sand and medium-sized gravel deposits. Within these frequently altering deposits, the Rhenish aquifer and the groundwater table are strongly impacted by compaction and consolidation of the various bedrock formations and recently by anthropogenic overprint, built-up activity, and climate change [87–89].

The Rhenish aquifer cannot be considered homogeneously distributed within the Quaternary gravel deposits and varies locally up to 14 m in depth and between 15 and 35 m in thickness [87]. Particularly the south-western part of the alluvial cone of the river Wiese shows a very high groundwater level anomaly with a maximum of less than 1 m below the modern surface. However, these surfaces are strongly biased by modern artificial transformation, built-up change, and sub-soil movements. Furthermore, the massive channelization activity and the increasing groundwater extraction for irrigation, industries, cooling, and private housing had a massive impact on groundwater temperature and eventually led to a drop of the groundwater table in the shallow gravelly aquifer during the past centuries and decades [90]. Most likely, this allows for the prediction of a higher groundwater table during Late Antiquity and the Early Middle Ages compared to the modern period. On the other hand, the drop in temperature and precipitation during the LALIA drought episode had a significant influence on the discharge of the river Rhine and the water table dropped accordingly [19].

**Geological regime and potential soil properties.** A generalized geological map visualizes the broad surface cover and can be used to estimate regional sub-surface conditions (**Fig 3A**). However, the terrain and the underlying bedrock vary on the very local level, which demands an individual model based on coring data and in situ samples.

3865 coring records were extracted from the WFS layer of the Basel administrative council. The point sample comprises information about the geological unit, surface altitude of the coring measure, and height of the bedrock surface. From the elevation columns, interpolations were carried out to visualize the surface rock height and to estimate the thickness of the aquifer (**Fig 3C and 3D**). Using the difference between the coring surface height and the bedrock surface measurements, the absolute thickness of the Quaternary gravel layer at Basel was calculated and interpolated using an inverse distance weighted interpolation (IDW) with a grid size of 5 m and the *gstat* package in R software (**Fig 3D**) [91, 92]. From the coring data, geological information of the underlying bedrock was extracted and frequencies > 5 were considered for the creation of a local geological map. 'Unknown' geological information was deleted from the data. Of the originally 35 different geological characteristics covering 3865 points, 17 were considered for classification with a total number of 3343 data points. However, the units are described as chronological periods or geological units, which results in a mixed categorisation table. Voronoi polygons were created and cropped to the extent of the Basel administration boundaries, which is the spatial extent of the coring dataset. The polygons visualize the geological information of the bedrock surface, covered by Holocene gravel and sand deposits (**Fig 3B**).

Low resolution geological maps mask the local geological and sedimentological heterogeneity. Extensive Pleistocene and Holocene gravel deposits as displayed in Fig 3A dominate the surface deposits of the river Rhine and the tributary floodplains. However, thickness of the aquifer and the underlying bedrock play a major role in groundwater circulation, affecting the river Rhine run-off behavior and discharge volume. The western part of the local bedrock at Basel is dominated by calcareous sandstones (Meletta) that merge into a heterogeneous mixture of yellowish, crumbly sandstone (Elsässer Molasse) and marl deposits and eventually into the so-called Tüllinger Schichten (verbatim Tüllinger units). The entire stratigraphy witnessed a mix of saline conditions during the Oligocene where marine sediments (Meletta units), followed by brackish (molasse) and limnic conditions (Tüllinger units, freshwater limestone) deposited [93]. The stratigraphy, however, is not planar but cut by the river Rhine palaeochannels to the west that forms a ridge facing northeast. To the southwest, the bedrock slope gradient increases continuously up to 35 m. With rising absolute elevation of the bedrock surface, the groundwater level increases accordingly and runs parallel to the former cut-out bedrock formation (**Fig 2C**). However, the bedrock surface slopes stronger than the current surface,

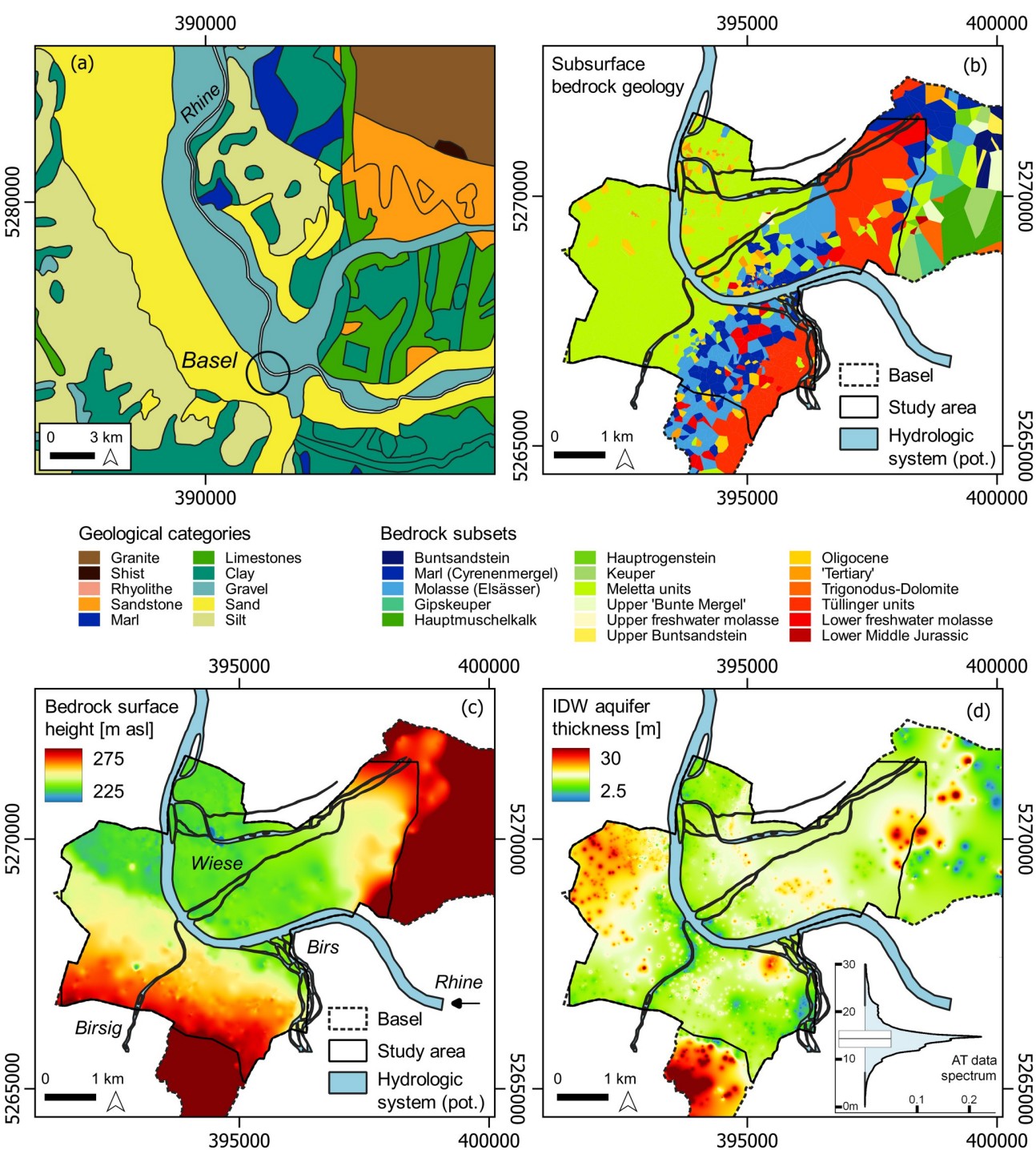

**Fig 3. Geological and hydrogeological conditions in the study area at Basel.** (a) regional, large-scale geological near-surface units; (b) locally estimated geological coring samples visualized as Voronoi polygons of the underlying bedrock surface (covered by Holocene gravel deposits); (c) multilevel b-spline interpolation of the underlying bedrock surface absolute height; (d) IDW interpolated aquifer thickness (Holocene gravel deposits) at Basel based on absolute coring height samples; boxplot and density curve of the data frequency. Images are for illustrative purpose only. Data source Fig 3 (a): geological map of Germany derived from INSPIRE under Creative Commons Attribution 4.0 International (CC BY 4.0) licence (https://download. bgr.de/bgr/Geologie/GK1000-INSPIRE/gml/GK1000-INSPIRE.zip, last accessed 3rd of November 2022).

which emphasizes geometry of the Holocene gravel deposits in the north-western part of Basel. Consequently, the aquifer thickness increases over 20 m with decreasing bedrock height.

The eastern bank of the river Rhine is characterized by a swift transition of Meletta sandstones to porous molasse and Buntsandstein formations. The bedrock height is about 238–240 m asl., and the aquifer thickness drops to about 13–15 m, which corresponds very well with the reconstructed pre-modern DEM (**Fig 2B**). Particularly the thin aquifer at the transition to the impermeable Meletta units [93] can be considered highly vulnerable to a rapid increase in groundwater circulation after heavy rainfall events and snow-melt in the Black Forest and the river Rhine catchment. The feedback between poor water storage capacity of the eastern Black Forest Mesozoic deck sediments, the confluence situation of the river Rhine and the river Wiese, and the low absorbing capacity make this region particularly liable to upwelling groundwater, flooding, and waterlogging conditions.

**Climatic conditions.** Central European climate and particularly the regional climatic conditions at Basel are influenced by the westerlies and the channelling effects of the so-called Belfort Gap, which opens up between the Vosges Mountains in the northwest and the Jura to the south. Consequently, the climate can be considered oceanic, with mild winters and warm summers, according to the Köppen Cfb climate zone [94]. The region is furthermore liable to pronounced heat waves [95]. In historical context, the climatic conditions between 100 AD and 800 AD were supposed to be relatively stable–compared to the large frequency oscillations of the late Holocene [96]. However, as the authors point out, socio-cultural and environmental systems that are located on the margins of stability are liable and sensitive to slight changes in the ecosystem functionalities, including temperature and precipitation regime. Large administrative bodies with centralized infrastructure and market-oriented, transnational economy like the Roman trade networks are more vulnerable to changes at the climatic peripheries, such as droughts in northern Africa or Sicily, or extensive flooding along the rivers Danube and Rhine, affecting crop production as much as transportation maintenance. Considering the climatic regime of the URA, subsistence land-use strategies could have played an important role for local farming communities, including short-term adaptation measures, such as drainage during excessively wet and irrigation following dry spells. However, long-term climate oscillations with subsequent years of harvest loss caused by droughts or constantly wet conditions can rapidly push the resilience of small-scale crop systems to a critical state.

Recent developments in dendrochronological research and results from ice-core proxy analyses have emphasized the drop in temperature that occurred during the second half of the 5th and the 7th century AD–labelled as the Late Antique Little Ice Age (LALIA) [97]. Fig 4 shows the climate variability over the period 100–800 AD based on data provided by Paul Krusic and Ulf Büntgen et al. (2021; 2011). Climate variability reconstruction is based on the (self-calibrating) Palmer Drought Severity Index (scPDSI/PDSI) [98] spanning annually resolved data composites for June, July, and August (JJA). For comparison reason, the reconstructed temperature anomalies (T[˚C]) and precipitation totals (P[mm]) are plotted with a correction factor for P (P = P[mm]/50). The Central European PDSI [17] emphasizes the persistent drought episode between approximately 425 and 620 AD that resulted from a drop in temperature and particularly in precipitation totals (LALIA drought period). A second PDSI time series was extracted for the region at Basel at a 0.25˚ grid cell resolution (Lat 47.5˚-47.75˚ x Lon 7.5˚-7.75˚). The data is based on the Old World Drought Atlas (OWDA) [16] and was extracted for the period 100–800 AD and equally plotted with a loess smoothing parameter of 0.1 (**Fig 4**).

Within this period, a sequence of volcanic eruptions has further dramatically impacted the Northern Hemisphere climate and eventually the vegetation response, land-use opportunities, and socio-cultural conditions [97, 99, 100]. The eruption series is supposed to have started

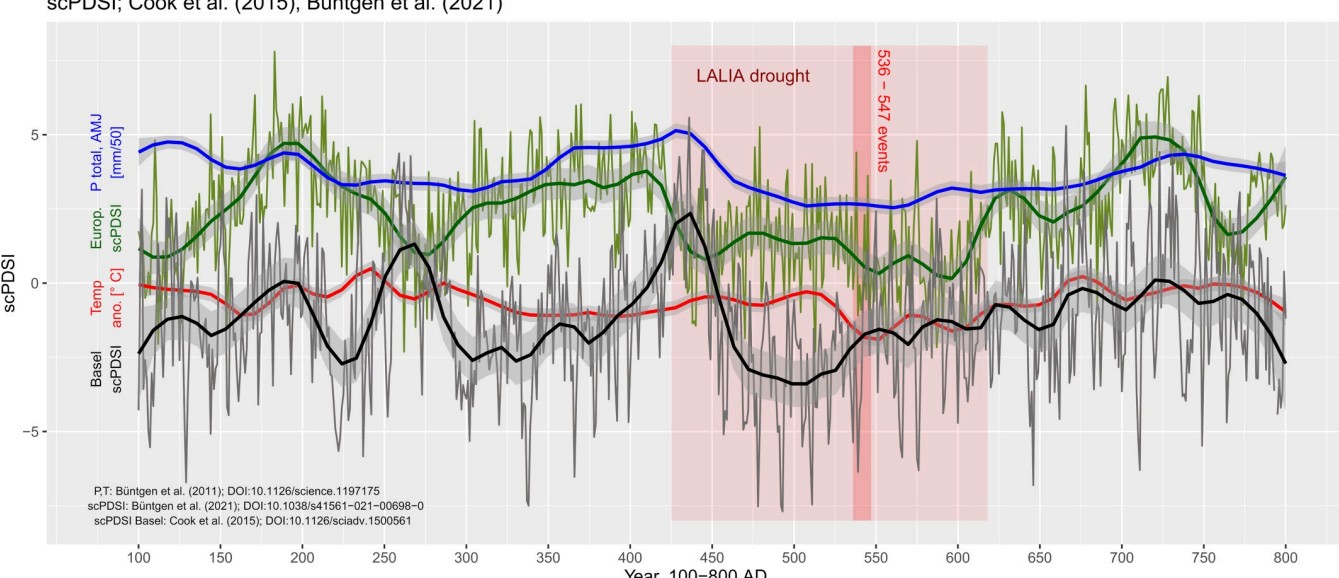

**Fig 4. Climate anomalies during 100–800 AD.** Green curve shows the Central European scPDSI reconstruction based on [17] and the black curve is the scPDSI at Basel [16], both lines are smoothed with a loess smoothing parameter of 0.1. Blue and red lines are equally smoothed precipitation totals (mm/50) and temperature anomalies (˚C) based on [18]. The Late Antique Little Ice Age (LALIA) is highlighted in red; the 536–547 AD volcanic events are marked by a red bar.

around spring 536 AD and most likely with an event in the Northern Hemisphere that exaggerated previously and later recorded eruptions (such as the Tambora eruption in 1815). It emitted a massive amount of dust particles into the higher atmosphere, which reduced solar radiation [101]. The acidic dust that was deposited in the Northern hemisphere, however, has been dated by Larsen *et al.* to around 533–534 ±2 years, which provides evidence for a potential response time of the tree growth in the northern hemisphere following the eruption and the dust transport across the equator [102].

Considering the regional context of the URA and the impact of the climate oscillations, the drop in temperature could probably have played a minor role for the maintenance of subsistence crop production and cattle breeding among small-scale farming groups. However, the significant drop in precipitation totals could have triggered local to regional adaptation mechanisms to acknowledge the decrease in groundwater availability and fresh water supply from the catchment areas of the Jura mountains and the Black Forest. Particularly the lower parts of the floodplain that are predominantly built by free draining gravel deposits from the river Wiese and the river Rhine do not provide a pronounced water storage capacity and are thus strongly liable to in-depth drying up. Local clayey infills on the contrary that show waterlogged conditions during wet spells can only be used as pastures due to their unfavorable topsoil compaction. Loess-covered areas on top of sandy plateaus that are slightly elevated over the floodplain farther north of Basel, where the river Rhine run-off characteristics shift to an anabranching system, provide fertile soil compositions but lack the water storage capacity in combination with lower groundwater table–making it particularly difficult to maintain irrigation of plants with very shallow root penetration. A potential technical solution to sustain crop production is the installation of irrigation channels produced by the continuous slope of the alluvial cone of the river Wiese. The low run-off velocity after entering the floodplain favours the construction of channels that can easily irrigate significant parts of the floodplain.

## Multiannual landscape vulnerability model

Using the reconstructed environmental data and the estimated run-off characteristics, a land-use vulnerability model was constructed that takes into account the geological conditions, aquifer thickness, run-off, and PDSI as well as topographic features such as slope gradient, elevation, and distance to the estimated hydrologic system at Basel. In the following section, a detailed description about the model set-up is presented, including an estimation of the river Rhine discharge during Late Antiquity and the Early Middle Ages. Eventually, the model can be resolved at annual scale or multiannual periods to estimate the landscape potential for agricultural crop production, pastures, and settlement spots.

**River rhine discharge estimation.** Run-off characteristics of the river Rhine for the period 100–800 AD were estimated using comparison run-off data from the sample period 1869–2012 (provided by the Bundesamt für Umwelt BAFU, Abteilung Hydrologie, Switzerland) and the PDSI within the catchment of the river Rhine (OWDA, Old World Drought Atlas, http://drought.memphis.edu/OWDA/, last accessed 04[th] of May 2022 [16]). From the data pixel-wise time series were extracted that cover the catchment area upstream of Basel (78.353,79 km$^2$) and mean values of the PDSI grid were calculated. The PDSI is based on June-August (JJA) composites and the run-off data was composed as mean values of three-month intervals to estimate synchronicity of the time series at variable periods across the year (MAM, JJA, SON, DJF (including JF+1)).

First, the time series were plotted over the maximum series length (1869–2012) to visualize and estimate potential correlation, coherence, and lags in the data (Fig 5). At first sight, the data seems to show synchronicity at least at specific periods, however, there are fluctuations at

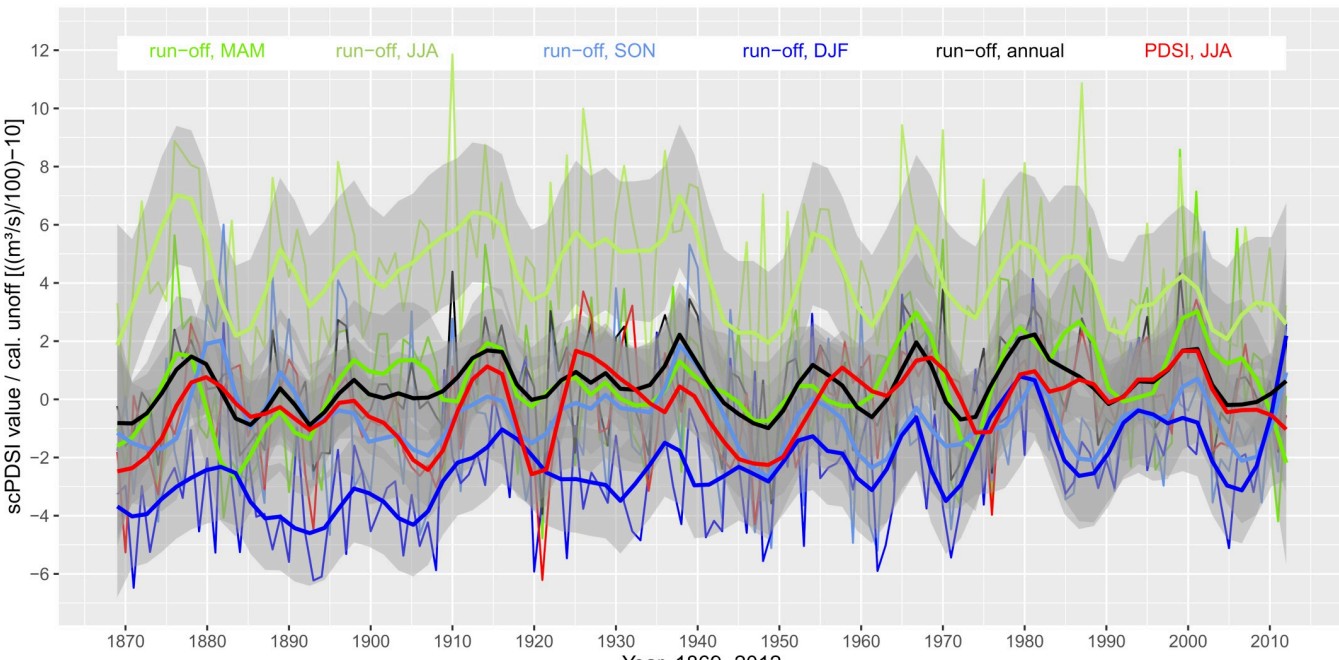

**Fig 5. Calibration data for river Rhine run-off at Basel.** Seasonal composites and annual mean river Rhine run-off characteristics at Basel, Rheinhalle, over the period 1869–2012 compared to mean PDSI in the river catchment (run-off adjusted as ((m$^3$/s)/100)-10; annual mean composites of MAM (March-May), JJA (June-August), SON (September-November), DJF (December, January (yr+1), February (yr+1); loess smoothing parameter = 0.1; grey shade = 95% confidence level).

variable scale at the seasonal level. The PDSI and the run-off data were analysed for coherence using Fourier analysis to estimate in-phase or out-of-phase behaviour over a particular period and to highlight returning patterns and lags in the data, basically referring to periodic phenomena in time series [103]. The frequencies of the time series were analysed for synchronicity at particular periods. In general, that is the relationship between two time series and their periodic phenomena, a statistic that is referred to as *coherence*. Coherence is defined as the square of the cross spectrum normalized by the individual power spectra and measures the cross-correlation of two time series as a function of frequency [104]. The R-package *waveletComp* [105] was used to analyse the frequency structure of the bivariate time series based on the so-called *Morlet wavelet* that detects continuous variations in signal periodicity during specific temporal periods [106–109]. The simulations produce a heat map of the cross wavelet power spectrum that emphasizes the periods (logarithmic scale) that are important for both time series at a specific time [103]. Horizontal arrows that point to the right show the in-phase relationship at the respective period and arrows pointing to the left emphasize anti-phase behavior with white contour lines around the arrows indicating high significance levels. The power spectrogram shows the frequency spectrum changes over a given time period and the coherence between the time series scPDSI over run-off. The vertical axis is the Fourier period, and the horizontal axis shows the time steps of the period 1869–2012.

In the period (frequency range) between 2 and 8 years, increased signals were recorded in the later 19[th] century, from 1910 to 1960, and during the late 20[th] and early 21[st] century (**Fig 6A**, left part). A plot of the time-averaged cross-wavelet power can be generated that shows the average power as a curve with 0.01 significance levels marked as red rectangles (**Fig 6B**, middle part). In this case, the average power decreases towards 1, between 2 and 4, and after 8 years with significantly strong ranges between 4–8 years period. Eventually, the wavelet coherence can be plotted (**Fig 6C**, right part). According to the authors of *waveletComp*, "the advantage of wavelet coherence over wavelet power is that it shows statistical significance only in areas where the series involved actually share significant periods" [109]. However, from the coherence plot, no inter-series synchronicity can be detected that would emphasize a causal relationship between PDSI causing run-off behaviour at variable time lag across the comparison period of 144 year.

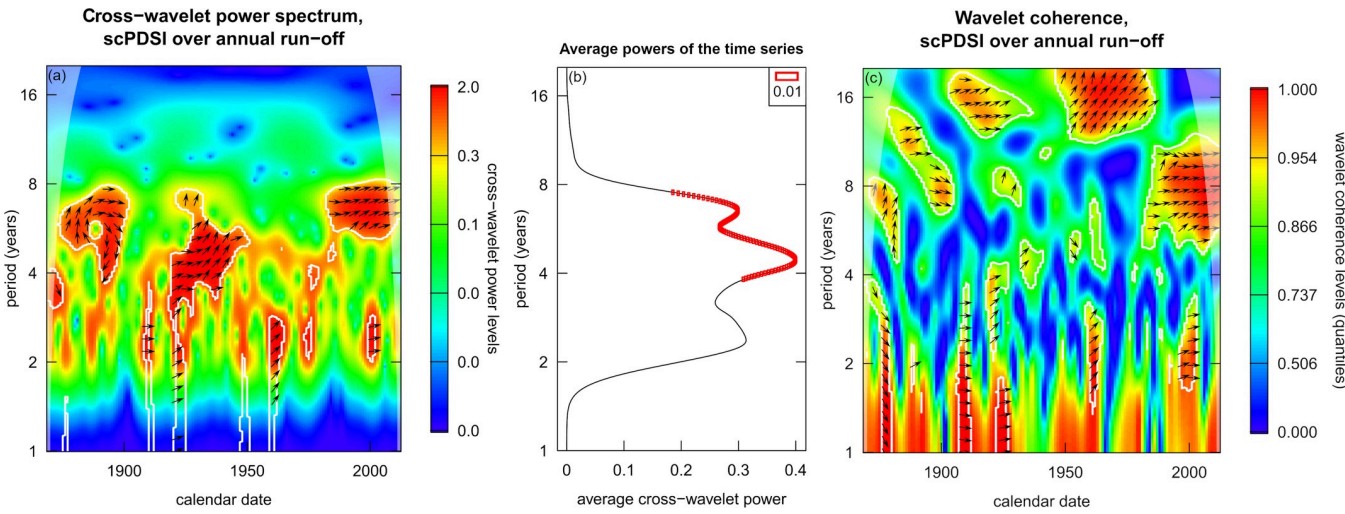

**Fig 6.** Plot of the wavelet analysis power spectrum (a), average powers of the time series (b), and coherence (c) for annual run-off at Basel and mean PDSI in the catchment over the period 1869–2012. Time-averaged cross-wavelet power plot with significance level of 1% (p-value 0.01); power spectrum window.size.s = ¼, coherence window.size.s = 1; nsim = 1000.

**Table 2. P-values of the Granger test for causality between PDSI (C) and seasonal annual run-off (T) using variable lags between 0 and 11 years.**

| | MAM | | JJA | | SON | | DJF | | annual | |
|---|---|---|---|---|---|---|---|---|---|---|
| lag | C→T | T→C | C→T | T→C | C→T | T→C | C→T | T→C | C→T | T→C |
| 0 | 0.3754 | 0.2438 | 8.09E-01 | 0.2438 | 0.6033 | 0.3205 | 0.3442 | 0.0005892 | 0.8392 | 0.2598 |
| 1 | 0.278 | 0.8104 | 2.98E-01 | 0.1004 | 0.1866 | 0.3779 | 0.8863 | 0.0002552 | 0.4797 | 0.3376 |
| 2 | 0.4572 | 0.2712 | 4.11E-01 | 0.5625 | 0.276 | 0.04819 | 0.1637 | 0.004067 | 0.4472 | 0.0782 |
| 3 | 0.7218 | 0.02793 | 1.65E-01 | 0.3632 | 0.9506 | 0.3656 | 0.672 | 0.005584 | 0.4778 | 0.09732 |
| 4 | 0.9037 | 0.008377 | 2.93E-01 | 0.5195 | 0.4675 | 0.2574 | 0.43 | 0.001597 | 0.648 | 0.02715 |
| 5 | 0.4193 | 0.08282 | 6.71E-01 | 0.9486 | 0.1469 | 0.417 | 0.8316 | 0.009594 | 0.9617 | 0.06796 |
| 6 | 0.339 | 0.6192 | 2.97E-01 | 0.0367 | 0.5152 | 0.09202 | 0.5862 | 0.0004386 | 0.9349 | 0.3341 |
| 7 | 0.2362 | 0.5031 | 9.36E-01 | 0.5867 | 0.2703 | 0.03536 | 0.8367 | 0.000371 | 0.8733 | 0.05549 |
| 8 | 0.3349 | 0.6729 | 4.30E-01 | 0.347 | 0.219 | 0.7208 | 0.8399 | 0.0004465 | 0.9872 | 0.3143 |
| 9 | 0.5463 | 0.2003 | 0.1436 | 0.02807 | 0.126 | 0.07068 | 0.6917 | 0.0001136 | 0.9867 | 0.1133 |
| 10 | 0.3906 | 0.5561 | 0.1815 | 0.7398 | 0.655 | 0.0276 | 0.9632 | 0.0249 | 0.8547 | 0.1108 |
| 11 | 0.7176 | 0.9132 | 0.1777 | 0.001952 | 0.6364 | 0.05427 | 0.6112 | 0.0003327 | 0.6514 | 0.6044 |

Consequently, further tests for causality or correlation are needed to identify the phase differences and lags in the datasets. First, a *Granger test* for causality was conducted with the 0-Hypothesis (*H0*) that one variable does not predict the other. It basically builds on the theory that a series *xi* is not considered causal to a series *xj* if using the history of series *xi* does not reduce the variance of the prediction of series *xj* [110]. In other words, does the past of variable *X* help improve the prediction of future values of *Y* (Granger causality) [111, 112]?. However, according to the review of Granger causality described by Shojaie and Fox (2022), the requirements of the test were manifold and hardly met in real world systems, including stationarity of the data [113]. Granger causality was operationalized in form of fitted VAR (Vector Autoregression) models, "simple mathematical models in which the value of a variable at a particular time is modelled as a (linear) weighted sum of its own past (usually over a number of discrete time-steps), and of the past of a set of other variables" [114]. The models can then be adjusted for phase differences using specific lags. For this article, lags between 1 and 11 years were calculated for PDSI predicting run-off and vice versa (**Table 2**), which show that *H0* cannot be rejected in the Granger test and that PDSI does not cause run-off. However, DJF run-off significantly causes PDSI across all lags (see **Fig 7**).

Eventually, the seasonal composites and the PDSI drought time series were analysed using a linear regression model. The regression model can be used to predict a variable on the basis of predictor variables ($X_1, \ldots X_n$). Model performance can be tested by evaluating a combination of different predictor variables that best predict another. Particular requirements for the model fit are normal distribution and homoscedasticity of the residuals [115]. A simple linear model has the equation:

$$Y = \text{ß}_0 + \text{ß}_1 * X$$

which reads *Y* equals $\text{ß}_1$ times *X*, plus a constant $\text{ß}_0$, where $\text{ß}_0$ is the intercept and $\text{ß}_1$ the estimated coefficient of *X*. *Y* is the response variable (target vector) and *X* the predictor variable (feature vector) [116]. In this case, the PDSI catchment mean values (JJA) from 1869–2012 (= *X*) were used to predict the run-off values at Basel Rheinhalle (= *Y*). We used several monthly and seasonal combinations of run-off to be best predicted by PDSI. Model performance can be traced with the AIC value (*Akaike's Information Criteria*), which is a best model selection method with the premise of the best model minimizing an expected discrepancy (low AIC values indicate best model performance) [117, 118]. The p-value of the model further indicates

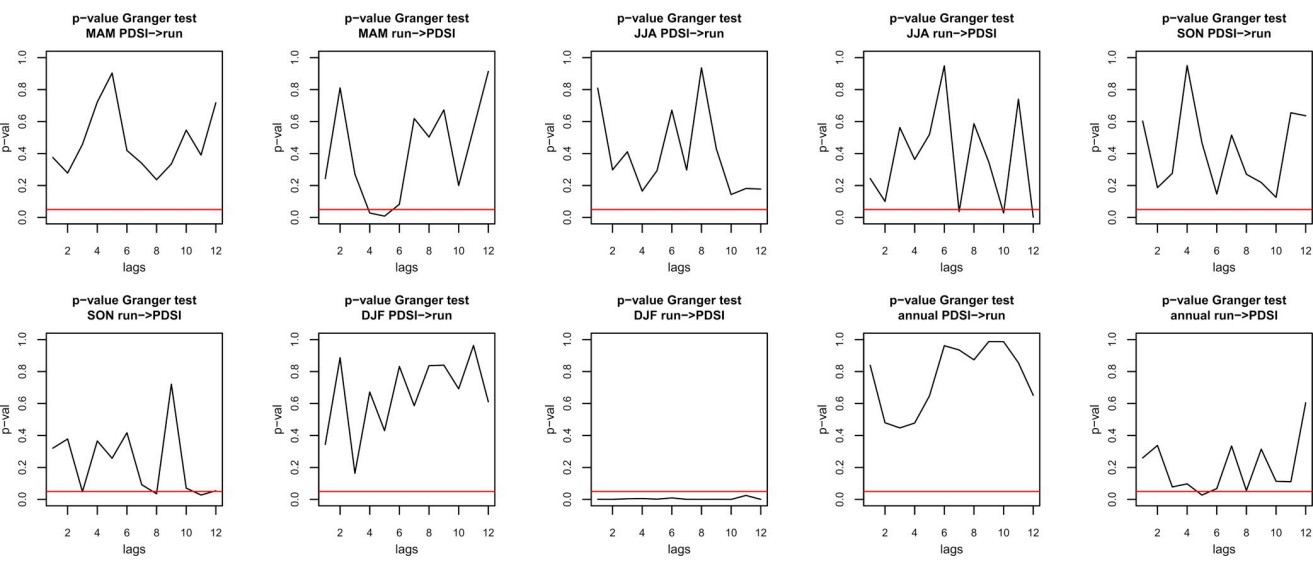

**Fig 7. P-values of the Granger test for causality using lags between 0 and 11 years.** PDSI cause run-off (PDSI->run) and run-off cause PDSI (run->PDSI). Significance level of 5% is marked with a horizontal red line (see Table 2).

whether *X* is significantly associated with changes in *Y*, and the coefficient is the estimate that describes the relationship between the predictor variable and the response. From the residuals, which is the difference between the observed values and the fitted values [116], we can derive how well the model fits the data. The residuals should be distributed symmetrically around the value 0. The normal distribution of the residuals can be tested using the *Shapiro-Wilk's test for normality* that has the 0-Hypthesis of normal distribution of the data [119]. A second constraint of the linear model performance is homoscedasticity of the residuals, that means that the variance is equal across the data. We can use the *Goldfeld-Quandt test*, which has the 0-Hypothesis that the residuals are homoscedastic [120].

The linear model that best predicts run-off behavior from PDSI at Basel has *Y* = run-off annual means and *X* = PDSI catchment mean composites of June-August values [16]. The model residuals range from -383.72 to 399.39 with an intercept estimate of 1053.728 and PDSI coefficient of 51.847. There is very significant probability that the linear model of PDSI predicts run-off (p-value = 5.498e-13) better than the intercept-only model (F-statistics have the value 63.08, R-squared = 0.3076). The AIC of the model is 1831.121, compared to 1969.17 (MAM), 1981.96 (JJA), 1980.85 (SON), and 1973.24 (DJF). The residuals are normally distributed with *p-value* = 0.6462 significance and heteroscedasticity can be rejected with *p-value* = 0.1825 significance. Outliers can be observed in the data using the *Mahalanobis distance*, which measures the distance between a point and the distribution (Fig 8). From the analysis, only a few outliers could be observed. Eventually, the regression model supports the hypothesis that the reconstructed PDSI can be used as indicator for river Rhine run-off behavior at Basel during the period 100–800 AD.

At Basel, the tributaries Wiese and Birs enter the river Rhine from the south and the north respectively. Both river run-off characteristics are available for the period 1917–2012 (Birs) and 1933–2012 (Wiese). Each catchment is represented by a unique PDSI value and the above-described method has been applied to analyse the relationship between PDSI and run-off in each catchment. Both results confirm the PDSI as indicator of run-off behavior, which further strengthens the model due to the very different micro-climatic conditions of the river Wiese

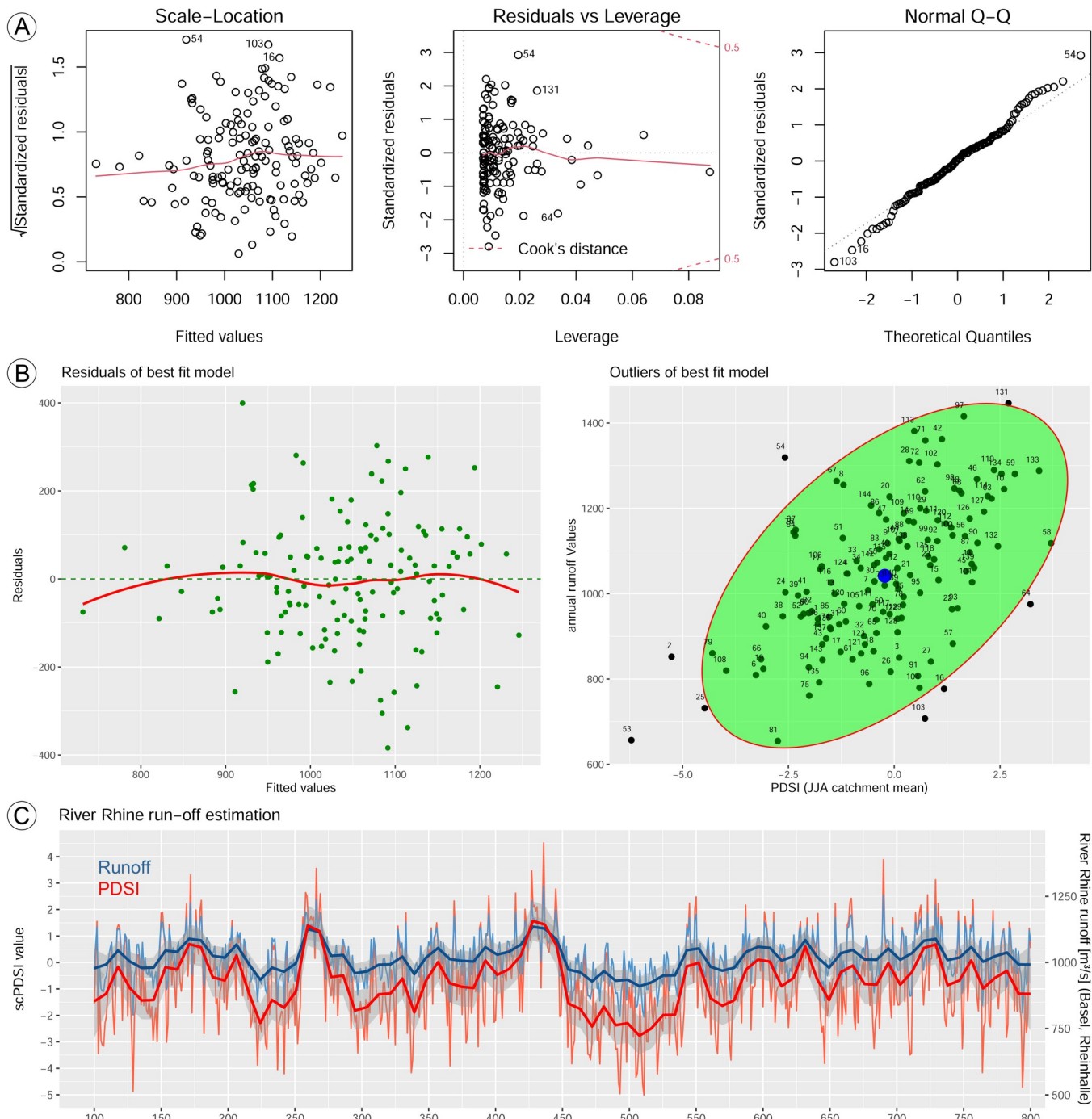

**Fig 8. Model performance of PDSI predicting annual run-off at Basel and run-off estimation for the period 100–800 AD.** (A) Model output, (B) residuals plot and Mahalanobis outliers; (C) estimated run-off behavior based on the predictor variables from the linear model.

catchment in the southern Black Forest and the river Birs catchment that stretches south towards the sub-alpine Jura Mountains.

**Model set-up.** Building on the discharge estimation, the annual run-off is used as coefficient that includes climatic parameters such as precipitation, snow-melt regime in the

catchment, and water storage capacity of the aquifer. Similar approaches have recently been developed for example by Hamer and Knitter (2018), who used fuzzy approaches to estimate land-use and location patterns in different areas and chronological periods [13, 121–123]. For Basel, the above-mentioned parameters were incorporated as raster data into the model and the distance of each cell to the closest river system was calculated using QGIS. First, a 5x5 m grid and centroids were generated, which were cropped by the hydrologic system to create a high-resolution point layer that covers all cells outside the river system polygons. The outlines of the polygons were converted to points and a point-based distance calculation of all centroids to the closest hydrologic margin point was calculated. Eventually, a multilevel b-spline interpolation was used to create a 5x5 m raster that assigns each cell the distance to the nearest water body. The hydrologic system itself was given the value 0. All input raster were cropped to the maximum extent of the smallest raster dataset to create equal extents for the analysis. Each raster was re-scaled to 0–1 using $X = \frac{X}{X_{Max}}$ and the model was created with the estimated run-off as coefficient for the hydrologic factors. The coefficient was adjusted using squared inverted and rescaled values between 0 and 1 to increase the power of climatic variability over the period and reads as

$$X_{Runoff} = \left( \frac{\frac{1}{X_{Runoff}}}{X_{MaxRunoff}} \right)^2$$

A second coefficient determines the topographic variables and the distance to the hydrologic system. The PDSI at Basel was used as drought coefficient. First, the values were re-scaled to positive values using the lowest PDSI as factor. The simple equation reads as

$$X_{Pos} = X_{PDSI} + X_{Min\ PDSI}$$

The data was then rescaled to range from 0 to 1 and inverted to make low values represent high PDSI values (wet years) and vice versa. The square root of the fraction reduces the impact of very low values during extremely humid periods and reads as

$$X_{SQRT} = \sqrt{1 - \frac{X_{Pos}}{X_{Max\ Pos}}}$$

Eventually, a simple moving average (SMA) of the 5 previous years has been applied to the data to integrate the effects of previous drought or wet spells at Basel. In a simple form, this reads as

$$X_{SMA\ (t)} = \frac{X_{SQRT\ (t-5)} + X_{SQRT\ (t-4)} + X_{SQRT\ (t-3)} + X_{SQRT\ (t-2)} + X_{SQRT\ (t-1)} + X_{SQRT\ (t)}}{5}$$

Eventually, we can construct the vulnerability model as

$$Y_{Vul} = ((X_{Elev} + X_{Slope} + X_{Dist}) * X_{SMA(t)}) + ((X_{GWater} + X_{AQ}) * X_{Runoff})$$

where $Y_{Vul}$ is an index of how liable the research area is to flooding, rising groundwater, and waterlogged soil conditions during and after heavy rainfall and seasonal run-off maxima. $X_{Elev}$ is the rescaled digital elevation model with 5 m resolution, $X_{Slope}$ is the reclassified slope with values ranging 0–1, and $X_{Dist}$ is the distance to the reconstructed hydrologic system. $X_{GWater}$ is the groundwater table interpolation and $X_{AQ}$ is the aquifer depth in the study area. High values show high agricultural and settlement security and low values indicate decreased suitability. The model is furthermore an indicator for general agricultural potential during wet and dry

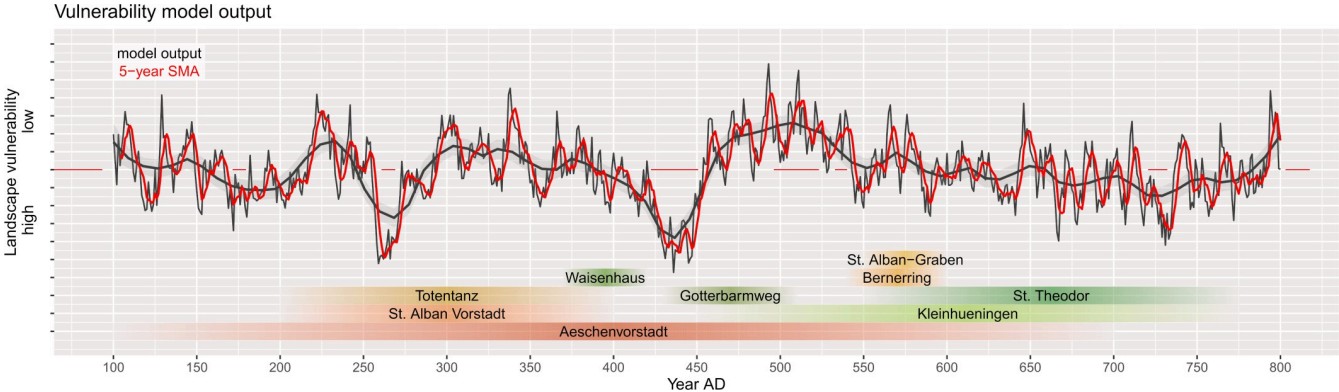

**Fig 9. Model output and occupation of the late antique and early medieval graveyards at Basel.** The black line represents the annual average model output (landscape vulnerability) with a loess smoothing parameter of 0.1. The red curve is a previous 5-year simple moving average (SMA). Graveyards are highlighted in green (northern bank of the river Rhine and red (southern bank of the river Rhine).

spells, for example during the dry LALIA. The model puts out annual raster that can be analysed for multiannual mean vulnerability using a moving window average (see **Fig 9** with a previous 5-year moving window). For visualization of average raster, 50-year splines provide a useful chronological differentiation to estimate land-use suitability and potential settlement locations that match the chronological classification of the early medieval typochronology (**Fig 10**).

## Results and discussion

River Rhine run-off estimates and PDSI are strongly connected through the model coefficients. However, the variability shows that during peaks of PDSI wet spells, the river run-off response is rapidly increasing which highlights the direct influence of precipitation in the catchment. Conversely, during prolonged drought periods, the run-off is less drastically affected but drops significantly. There is evident climatic and landscape variability across the study area during the period 100–800 AD, which is mirrored in strong oscillation of the reconstructed PDSI values at Basel [16], the general climate variability across Central Europe [17, 18], and the surface model that integrates multiple factors such as local geological and topographic features. Climatic variation had a significant impact on the local surface und sub-surface suitability for agricultural development and settlement strategies, which most likely triggered constant relocation of houses and fields as a result of periods characterized by increasingly dry or humid conditions. Soil humidity is a particularly important issue for early medieval settlements in Central and Western Europe, where pit-houses and silos belong to the typical features of a rural settlement [41, 124]. In this context, shifting settlements are a phenomenon commonly observed for the Early Middle Ages, in which frequent reconstruction of wooden structures maintain a specific core area but allow for widespread settlement patterns [4, 125].

### The long late antiquity

During the late Roman period, a first pronounced disturbance in climate stability occurred between 200 and 250 AD, coupled with rising temperatures and a decrease in precipitation. Consequently, the first half of the 3rd century AD is dominated by rather dry conditions, which favoured prolonged drought periods. This changed rapidly between 250 and 275 AD, a period that was accompanied by a significant drop in temperature [126]. This period is further characterized by a volcanic eruption around 250 AD that increased the sulphate content in the

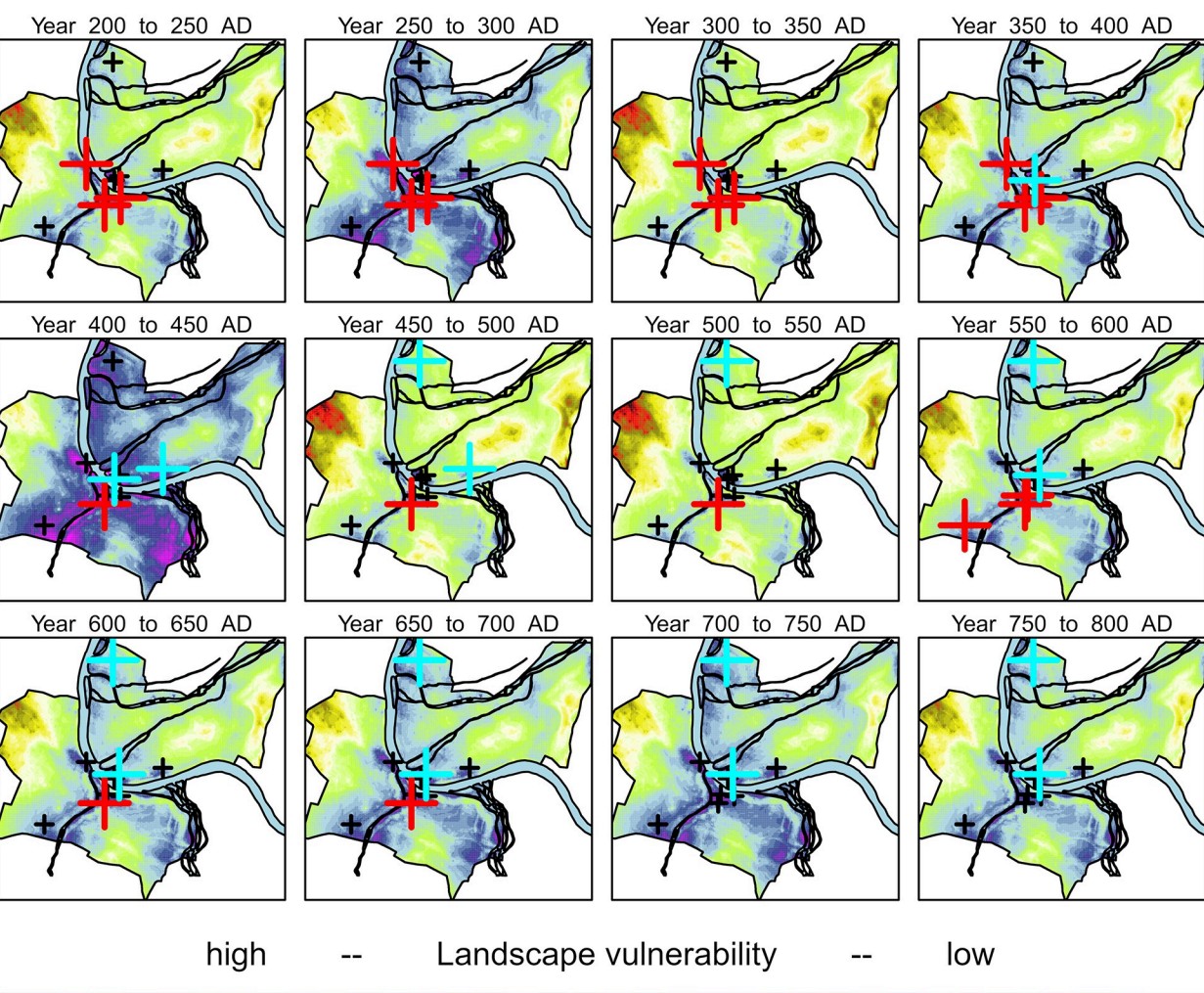

**Fig 10. Model output of 50-year averages across Basel between 200 and 800 AD.** Chronological occupation of the graveyards are given in red (southern bank of the river Rhine) and cyan (northern bank) (see **Fig 1**).

atmosphere [127] and triggered or at least contributed to the rapid drop in temperature. At Basel, this drop must have caused massive changes in precipitation patterns and run-off behaviour of the river Rhine and the tributaries at least during the third quarter of the 3rd century AD. As shown in Fig 4, a major wet spell occurred during 250 and 275 AD, which marked a peak in the late Roman period at Basel. This peak is not visible in the European PDSI reconstruction and the groundwater estimation [17, 19] and can point to regional environmental feedbacks rather than Northern Hemisphere climate patterns. The late Roman warm period as reported from Tegel *et al.* (2020) is further not clearly visible in the Basel PDSI and the river Rhine catchment PDSI values (Fig 4). Rather, the data emphasizes a prolonged moderate drought period accompanied by a temperature decline [18] after 275 AD.

It is noteworthy that these climatic changes correspond to the period at which a large part of southwestern Germany (known as *agri decumates*) was abandoned by the Roman

population [22]. The traditional explanation for the fall of the Upper Germanic-Rhaetian Limes refers to the low population density of this area leading to a loss of rentability, and the difficulty to protect this peripherical region against barbarian raids during the so-called 'Crisis of the Third Century' [30, 37, 128]. A deterioration of climatic conditions and its impact on the agricultural production could have played an important role in the equation. Across Europe, the 4th AD century did not show pronounced drought periods but rather constantly increasing precipitation records that peaked around 410 AD. At Basel, however, this period coincides with very dry conditions during the first half of the 4th century AD, replaced by a pronounced maximum in humidity exactly between 425 and 450 AD.

In Basel, the only archaeological records known for the 3rd and 4th centuries AD are concentrated in close vicinity to the late antique military facilities such as the *castrum* on the southern and the *munimentum* on the northern riverbank [2, 40]. When considering the lack of evidence for an intense settlement activity on the northern bank during this period, the even worse climatic conditions at Basel compared to Europe average might rather explain the low density of occupation in the region–principally reduced to the military facilities–instead of a massive arrival of Germanic groups who allegedly invaded or migrated to the area from the late 3rd century AD onwards [24, 37]. However, this situation may be biased by the current state of research. The funeral practices during this period are, for example, characterized by burials with no or hardly any grave goods. Because this was traditionally considered as a Roman tradition [2, 129, 130], and the northern riverside was abandoned by the Romans around 260 AD, the (quasi) empty burials in Basel-Kleinhüningen and Basel-Gotterbarmweg located beyond the limes were never interpreted as potentially Roman. Without radiocarbon dating, however, it is not possible to exclude the hypothesis that both graveyards already started in Late Antiquity. Some very scarce and still poorly understood late antique finds from the area around Basel-Kleinhüningen could support this hypothesis [131]. On the other hand, the lack of grave goods in late antique burials might also be related to the seemingly challenging living conditions leading to a reduced need to emphasize wealth in the funeral context. This could explain why the first half of the 5th century AD is so underrepresented in the archaeological record.

## The LALIA at Basel

The very wet conditions right before the drought spell during the LALIA caused a massive increase in river discharge volumes and must have locally caused strong flooding events in combination with upwelling groundwater in areas with a thin aquifer over impermeable bedrock. Eventually, significant parts of the available settlement areas at Basel would have shown high vulnerability to general wet surface and waterlogged conditions. Particularly on the northern riverbank, potential settlement spots would have been located on rather elevated areas with low-lying and thick aquifer in considerable distance to the river Rhine and the tributaries. Potential cropland was equally liable to extensive flooding, particularly during the early growing season and the snow melt period, which would have increased the risk of harvest loss. Large parts of the floodplain would be suitable only as pastures and agricultural production would have moved towards the central and higher parts of the alluvial fan, where a lower groundwater table prevails (**Figs 2 and 3**). The southern riverbank is characterized by stronger topographic variability in the range of the Münsterhügel that stretches along the southern riverbank. These bedrock outcrops show high settlement security throughout wet spells and were presumably continuously occupied. The surroundings, however, show an increase in groundwater height and a decrease in aquifer thickness, particularly amplified in close distance to the tributaries Birs and Birsig. During intense rainfall in the Jura Mountains, a rapid and strong

increase in river discharge would lead to backlog conditions and flooding of the lower lying floodplain areas. Crop production in this area would be highly vulnerable to extreme weather events and changes in rainfall patterns in the catchments of the rivers and not necessarily at Basel itself.

The drop in run-off during the LALIA and particularly between 450 and 550 AD is visualized in Figs 8 and 9. Compared to recent results by Tegel *et al.* (2020), which show reconstructed Upper Rhine groundwater levels based on tree-ring data, the onset of the drop in PDSI and the correlated run-off can be observed earlier in the study area [19]. The drought period that is characteristic for the LALIA had an impact in the river Rhine catchment area starting from 450 AD and lasting throughout the 5th and the first half of the 6th century AD. Drought conditions can be assumed for Basel only after 450 or even 475 AD and the local response to decreased precipitation is much more intense than the European trend (Fig 4).

At the latest around 450 AD, after the humid phase, the first reliable evidence for burial activity at Basel- Kleinhüningen and -Gotterbarmweg are attested. This preceded the administrative collapse of the Western Roman Empire, which suggests that the development of the settlement activity on each riverside was not impeded by the vicinity to the limes. At the same time, it is not possible to assess if the sudden increase in quality and quantity of grave goods was related to more prosperity or to social competition and the need to express or legitimate power in a politically unstable period [132, 133]. After 500 AD, the Western Roman Empire no longer existed, the area fell under Frankish administration, and the drought and cool climatic conditions that accompanied the LALIA already lasted for half a century. The settlement activity was now apparently reduced to the large cemeteries located on each riverside: Basel-Aeschenvorstadt and Basel-Kleinhüningen. The important number of burials without grave goods at both sites and the implied absence of a precise chronology do not allow to determine if there was a demographic decline during this period or not. But the archaeological record at least shows a continuity of activity at Basel. The number of lavish burials is reduced compared to the previous decades, though the diversity of cultural influences among the grave goods over the whole LALIA shows that at least some individuals were integrated into widespread socio-cultural and economic networks [32]. This suggests a certain degree of prosperity and stability despite cold and dry conditions.

## The late 6th and the 7th century AD

The second half of the 6th century AD can be considered a trend towards short-term climatic variability and a general trend in stability, which lasted throughout the 7th century AD (Fig 4). However, recent results contribute to the discussion about the actual century-long cooling during the LALIA and emphasize a rather decadal response to a series of volcanic eruptions, probably visible in the PDSI after 536 AD [134]. Regional socio-environmental transformations, however, cannot be compared equally across Europe, which makes it particularly difficult to estimate global trigger and local or regional response. Drought spells and flooding occurred simultaneously across Europe during the first half of the 6th century [135, 136] and local to regional adaptation to environmental changes followed frequent and rapid increase in vulnerability. The data used in this article shows slowly increasing temperature development after 550 AD, accompanied by an increase in precipitation or general humidity. On a decadal level, these variations fluctuate more frequently but with lower magnitude. However, this does not necessarily increase stability of socio-ecologic systems at the ultimate margins of climate sensitive areas, such as lower floodplains. Compared to dry periods (e.g., 450–550 AD), the later 6th and 7th century AD show increasingly vulnerable areas particularly along the tributaries and in the confluence zones with the river Rhine (Fig 10). Potential settlements would

therefore be located rather to the north-eastern part of the northern riverbank and the southern part of the southern riverbank as well as the Münsterhügel.

From the late 6th century onwards, the development of the burial practices including a decrease in the number of grave goods and a separation of some burials at the edge of the main burial grounds (Kleinhüningen and Aeschenvorstadt), along the roads, or on new, smaller places (Bernerring, St. Theodor), reflect the trends observed in Europe and may be more related to a new way of social representation than to a direct response to changing climatic conditions [70]. The lavish burials at Basel-Bernerring suggest both prosperity [66] and the continuity of extensive networks [32]. In this specific context, the strategic aspect of the position on the long-distance route connecting France with Italy seems to have played a prevailing role in the location choice compared to the environmental settings in the direct surroundings [32, 66]. This is supported by the location of other burial places further north and single graves in stone coffins further south along the same road. On the northern riverside, the burial place around present-day St. Theodor might have been already related to a church in the early stages of a Christianization, emphasizing the increasing power of the church as institution, even before the bishop seat was transferred to Basel [1, 34].

## Settlement dynamics

The Late Antiquity and Early Middle Ages were traditionally considered as a transition period between the Roman Empire and the medieval kingdoms. However, this study reveals an intensive, continuous, and highly dynamic settlement activity at Basel during the whole period. It is particularly noteworthy that at least two burial places were used without hiatus over several centuries: Basel-Aeschenvorstadt on the southern and Basel-Kleinhüningen on the northern riverbank. This illustrates a certain degree of stability in this area despite the location at the border of the Western Roman Empire and the considerable political and administrative changes described in the written sources. The locations of the settlements potentially shifted in the surrounding areas as a response to climatic and demographic changes, but these burial places apparently remained milestones in the local tradition, shaping the landscape over centuries. The settlement at Kleinhüningen has not been discovered yet, but regarding Aeschenvorstadt, there is a probability that the place was used by at least part of the people living on the Münsterhügel [2]. Especially when the *castrum* was used as a late Roman military station from the 3rd/4th to the late 5th century AD, it is not excluded that other communities were using the various late antique burial grounds at Aeschenvorstadt, St. Alban-Vorstadt, and Totentanz, and to some extent also at Waisenhaus. Even after the Münsterhügel lost its military function in the late 5th century AD, the place remained a centre of political, religious, and economic power [3], highlighting its role as a central place. In combination with a strategic and secure situation on a sloping hill, this could be a reason for the long-lasting continuity of the burial place at Basel-Aeschenvorstadt, despite the variability in climatic conditions.

On the other hand, some burial places were used for only a couple of decades, suggesting that there was a fluctuation in the settlement activity over time, with periods of concentration in the two main spots at Aeschenvorstadt and Kleinhüningen alternating with a period of more scattered occupation. It is hard to recognise a distinct correlation between the settlement and the climatic cycles, which further confirms that settlement activities are related to both environmental affordances and socio-political issues [9, 137]. Concerning Basel-Gotterbarmweg, it is for example striking that this community started burying their deceased at a new place just during a period of deteriorated climatic conditions. Does it mean that the local carrying capacity was reduced and that there was a need to settle in new areas to increase resilience? However, the prestige of the burials rather suggest prosperity. At the same time, it is

noteworthy that the burial activity at Basel-Gotterbarmweg also corresponds to the decades before and after the collapse of the Western Roman Empire, and that the burial ground is located close to a potential river crossing point. It is possible that the corresponding community was also involved in controlling this important connection route during this key period. A similar conclusion can be drawn for Basel-Bernerring, which community might have settled at this specific place after the Frankish conquest to control the long-distance route connecting France and Italy. In this case, however, the increase in landscape vulnerability at this specific location during the 7[th] century AD might have triggered the end of the burial–and potentially the settlement–activity.

The few individuals buried at the Antikenmuseum potentially show a similar pattern of separation related to a local deterioration of climatic conditions, but the archaeological context does not provide enough data to draw any conclusion regarding this small burial place. And again, the high quality and quantity of grave goods in two burials does not give the impression of a weakened economy either. As mentioned above, the development of the burial ground at St. Theodor might equally be more related to socio-political issues than to climatic conditions. Even though a large-scale amelioration might have triggered the European wide social changes. In any case the distance between the burial grounds and thus between a potential corresponding settlement is always large enough to enable each community to have a separate catchment area. But it is not excluded that several hamlets or scattered houses shared the same burial ground, which would increase the density of scattered buildings around each site. Except for the late antique burial places mentioned above, the considerable distance between the graveyards as well as the diversity in the burial practices and network relationships rather suggest that each burial ground was used by a different settlement community. The continuity of settlement, the evidence for high grave good quality over time, and the long-distance socio-political and economic networks [32] suggest a continuous prosperity of the area despite dramatic political changes such as the collapse of the Western Roman Empire and the Frankish (administrative) conquest.

It is nevertheless difficult to assess to what extent the river Rhine either before, during, or after its role as late Roman limes divided the area and kept both riversides apart. Significant differences in funeral practices and in cultural influences for at least some individual burials rather point towards separation. But recent studies also highlight the long-lasting continuity of late Roman tradition on both riverbanks [40] as well as the integration of each site at Basel into a cultural area spanning the region at Basel, Southwestern Germany, and the hinterland of the river Rhine [32]–demonstrating substantial interaction between both riversides and their complementary regions.

## Conclusions

Regionally diverse precipitation patterns and changes in temperature had a substantial impact on late antique and early medieval landscapes across Europe. This is particularly evident in climatically sensitive and environmentally vulnerable areas such as the high latitudes and altitudes and semi-arid regions, but also in areas liable to flooding, waterlogging, or near-surface soil desiccation. In this context, drying up under colder and drier conditions can trigger adaptation measures in terms of agricultural development and landcover transformations of previously unsuitable areas such as floodplains or seasonally flooded and waterlogged meadows. And climate fluctuation can trigger further landscape development, including shifts in settlement spots, agricultural cropland, or pastures. Here, we presented a comprehensive socio-environmental analyses to create a complex model of regional landscape vulnerability at Basel, Switzerland during the Late Antiquity and the Early Middle Ages. We highlighted land-use

suitability at 50-year splines based on annually resolved model outputs to visualize trends in potential settlement and cropland occupation. From the model, we derived that areas, which have been suitable for settlements and agricultural crop production during the so-called LALIA drought period have become unsuitable during shifts towards more humid and warmer conditions. These periods triggered an increase in river Rhine run-off, higher groundwater levels, and seasonal flooding after heavy rainfall and amplified snowmelt during late spring and early summer. Consequently, a shift towards low-lying parts of the floodplain was more secure during drought periods, which would have further enabled the establishment of a local burial ground in close proximity to a rural farming community. During more humid phases, these areas were unsuitable for settlements and particularly for burial grounds due to high groundwater levels and seasonal waterlogging of the near-surface subsoils. Such local constraints would be less appropriate during a contextualized burial choreography and would have particularly affected burial practices in winter due to frozen topsoil.

Considering the evolution of climatic conditions and the local response to changes in temperature, precipitation, and runoff, is key to understand settlement patterns in a local or regional archaeological model. At Basel, the settlement continuity and the permanent development of a particular burial activity on both sides of the river Rhine suggest the great adaptability of late antique early medieval societies to both climatic and political disturbances.

## Acknowledgments

We are grateful to Philippe de Smedt for helpful comments during the editing process. Furthermore, we would like to particularly thank Piraye Hacıgüzeller and Bert Groenewoudt for their very constructive suggestions and positive feedback on this article. This applies also to Paul Krusic and Ulf Büntgen from the University of Cambridge for discussion and assistance during the model set-up. We are further very grateful for the structural input and the very helpful assistance by Alfredo Cortell-Nicolau from the McDonald Institute for Archaeological Research at Cambridge. We would particularly like to thank Tamsin O'Connell, Enrico Crema, and all colleagues from the Department of Archaeology and the McDonald Institute for Archaeological Research at Cambridge for supporting our work during our fellowships. We would like to thank Nik Jauer and the whole team of the Eidg. Departement für Umwelt, Verkehr, Energie und Kommunikation UVEK, Bundesamt für Umwelt BAFU, Abteilung Hydrologie for providing gauge data at Basel.

## Author Contributions

**Conceptualization:** Michael Kempf, Margaux L. C. Depaermentier.

**Data curation:** Michael Kempf, Margaux L. C. Depaermentier.

**Formal analysis:** Michael Kempf.

**Investigation:** Michael Kempf, Margaux L. C. Depaermentier.

**Methodology:** Michael Kempf, Margaux L. C. Depaermentier.

**Project administration:** Michael Kempf, Margaux L. C. Depaermentier.

**Software:** Michael Kempf.

**Supervision:** Michael Kempf, Margaux L. C. Depaermentier.

**Validation:** Michael Kempf, Margaux L. C. Depaermentier.

**Visualization:** Michael Kempf.

**Writing – original draft:** Michael Kempf, Margaux L. C. Depaermentier.

**Writing – review & editing:** Michael Kempf, Margaux L. C. Depaermentier.

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
