## [Decision Letter · Decision Letter 0]

11 Oct 2022

PONE-D-22-18464Societal and environmental transformations in late antique and early medieval Basel, SwitzerlandPLOS ONE

Dear Dr. Kempf,

Thank you for submitting your manuscript to PLOS ONE. After careful consideration, we feel that it has merit but does not fully meet PLOS ONE’s publication criteria as it currently stands. Therefore, we invite you to submit a revised version of the manuscript that addresses the points raised during the review process. While only minor revisions are requested, particularly the comment by reviewer 2 on describing the model variables more fully should be addressed.

We look forward to receiving your revised manuscript.

Kind regards,

Philippe De Smedt

Academic Editor

PLOS ONE

Journal Requirements:

MK received funding from the Masaryk University (grant number CZ.02.2.69/0.0/0./18_053/0016952; Postdoc2MUNIm order number 21 0053) and from the CRC1266 ‘Scales of Transformation’ at the University of Kiel and the German Research Foundation (grant number 290391021). MLCD’s research at Basel was funded by the University of Basel and the Swiss National Science Foundation (SNSF). 

However, funding information should not appear in the Acknowledgments section or other areas of your manuscript. We will only publish funding information present in the Funding Statement section of the online submission form. 

MK received funding from the European Commission and the Masaryk University, Brno; Czech Republic, grant number. CZ.02.2.69/0.0/0./18_053/0016952; Postdoc2MUNIm order number 21 0053.

MK received funding from the German Research Foundation (DFG) and the CRC1266 Scales of Transformation, Kiel University, grant number: 290391021.

MLCD received funding from the University of Basel and the Swiss National Science Foundation (SNSF).

5. We note that Figures 1, 2 and 3 in your submission contain map images which may be copyrighted. All PLOS content is published under the Creative Commons Attribution License (CC BY 4.0), which means that the manuscript, images, and Supporting Information files will be freely available online, and any third party is permitted to access, download, copy, distribute, and use these materials in any way, even commercially, with proper attribution. For these reasons, we cannot publish previously copyrighted maps or satellite images created using proprietary data, such as Google software (Google Maps, Street View, and Earth). For more information, see our copyright guidelines: http://journals.plos.org/plosone/s/licenses-and-copyright.

a. You may seek permission from the original copyright holder of Figures 1, 2 and 3 to publish the content specifically under the CC BY 4.0 license.  

6. We note you have included a table to which you do not refer in the text of your manuscript. Please ensure that you refer to Table 1 and 2 in your text; if accepted, production will need this reference to link the reader to the Table.

Reviewers' comments:

Reviewer's Responses to Questions

**Comments to the Author**

1. Is the manuscript technically sound, and do the data support the conclusions?

Reviewer #1: Yes

Reviewer #2: Yes

2. Has the statistical analysis been performed appropriately and rigorously? 

Reviewer #1: I Don't Know

Reviewer #2: Yes

3. Have the authors made all data underlying the findings in their manuscript fully available?

Reviewer #1: Yes

Reviewer #2: Yes

4. Is the manuscript presented in an intelligible fashion and written in standard English?

Reviewer #1: Yes

Reviewer #2: Yes

5. Review Comments to the Author

Reviewer #1: This is an excellent and also innovative paper, although the introductory chapters 2.1.2 and 2.1.3. could have been written a bit more concise. The reliability and usefulness of environmental models does indeed to a large degree depend on the number of chosen variables, the comprehensiveness of the explanatory parameters, and (importantly!) the integration of socio-cultural decision-making into the model. The authors have clearly demonstrated that their approach (PDSI) has great potential to improve the spatio-temporal resolution of reconstructions concerning past human environment (and climate) interaction.

Reviewer #2: This article aims to reconstruct the development of rural settlements and land use patterns (i.e. crop production and pasture use) during the late antique and early medieval occupation of the Basel area.

The aim is to perform this reconstruction using information on topography, climatic conditions, soil properties, and ground and surface water resources.

The study succeeds in producing a series of important results for the period and area that trace the relationship between settlement development, crop production and pasture use, and drought, flooding, rising groundwater and waterlogged soil conditions.

It will require minor revisions, after which it will be ready for publication, I think.

Strengths:

- The article is well written, well structured.

- The methodological workflow based on freely available information online is a major strength of the research and sufficient detail is presented to replicate the research.

- The variables used in the study to build the “multiannual landscape vulnerability model”, namely graveyard and burial locations, topopgraphical and soil properties, climatic conditions and evidence for water resources, can be considered as adequately measured, despite the various assumptions made (including those related to water run-off).

- The consideration of the (Palmer Drought Severity Index) PDSI as an indicator of the runoff behavior used for the estimation of the Rhine discharge in section 2.4.1 seems correct.

- The model presented in section 2.4.2 is a good representation of the groundwater level and the aquifer thickness, as well as the topographical properties (i.e. elevation, slope, distance to hydrological systems such as the Rhine system) in the study areas where the probability of flooding, rising groundwater and waterlogged soil conditions are modelled.

Weaknesses:

The study has few weaknesses. It is transparent in its handling of the data, which is open and available, and in the assumptions it makes when processing the datasets and creating the “multiannual landscape vulnerability model”. I noticed two relatively minor issues here:

- The current title of the article, “Societal and environmental transformations in late antique and early medieval Basel, Switzerland”, does not accurately reflect the content. A more expressive title (perhaps an added subtitle?) should probably refer to the research focus on rural settlement development and land use patterns, and the modeling approach used.

- In section 2.4.2 it would be good to describe all variables used in the model. For example, at the moment v is not defined like many others.

Minor issues:

- Lines 69-71: There is something wrong with this sentence: “This is making is particularly difficult to trace past human activity spheres and to put them into a meaningful context under consideration of past environmental conditions.”

- Lines 96-99: This information is better included as end-/footnotes if possible: “(R environment: The R Project for Statistical Computing, https://www.r-project.org/about.html, last accessed, 28th of June 2022; QGIS, https://www.qgis.org/en/site/, last accessed, 28th of June 2022).”

- Lines 125 – 127: Check the use of “which” which may not be correct: “During the late 5th to the early 6th century AD, the region was included into the Frankish Kingdom, which administration and infrastructures still relied on Late Roman legacies [28–30].”

- Line 165: Change “know” to “known”

- Line 218: why is the number of burials (n) sometimes further specified as nLATEANTIQUITY here and sometimes not?

- Line 234: Museum of Antiquities of which city? Please specify

- Line 272: Put a space after “[55]”

- Line 341: Check the wording: ”the past 19th and 20th years”?

- Lines 361-362: Avoid using “harmonized” twice.

- Lines 512-515: Revise sentence: consider using “for” instead of “of” in “important role of local farming communities”, and best avoid using “during” twice.

- Line 979: “This illustrate” should become “This illustrates”

6. PLOS authors have the option to publish the peer review history of their article (what does this mean?). If published, this will include your full peer review and any attached files.

Reviewer #1: **Yes: **Dr. Bert Groenewoudt

Reviewer #2: **Yes: **Piraye Hacıgüzeller

---

## [Author Response · Author response to Decision Letter 0]

4 Nov 2022

Dear Reviewers,

Many thanks for the productive suggestions regarding our article! In the following, we want to emphasize the few changes to the original draft and answer the issues raised by the two reviewers.

We changed the title according to suggestions by Reviewer 2. We now refer to the modelling character of the paper and the settlement and land-use spatial analysis. 

We included a short description of the variables of the model (e.g. Xelev) in section 2.4.2. We hope that it becomes more clear now. We already solved all the minor issues raised by Reviewer 2. 

Overall, we are extremely grateful for the very positive feedback on the paper and particularly on the model. We are looking forward to further explore the run-off model and integrate a continental-wide approach in a future project! We included both reviewers into the acknowledgement section of the paper!!

Point-by-point response to reviewers:

Rev#1:

- Many thanks for this very positive review! We are delighted to see how the model, the workflow, and the results were appreciated! 

Rev#2:

- Again, many thanks for this equally positive and constructive review! The minor changes raised here will be listed and answered in the following:

- Title change: as described in the general comments above, we changed the title to better fit the paper. You were right that a too general title does not actually refer to the modelling approach and the case study!

- Variable description: we described the variables used in the model, like elevation, slope and others. We initially thought that they were self-describing but you were completely right that in formulas, this is not always the case! We hope that now it becomes cleared what are the input variables. Many thanks for this!

Minor issues:

- Sentence corrected

- Footnote included with description

- Sentence corrected

- Known corrected

- Former line 218: n(late antiquity) refers to the occupation time of Aeschenvorstadt, which runs over a long period, including also Early Medieval period burials. Here, we emphasize the late antique occupation. 

- Museum of Basel, corrected in the MS

- Space put

- Wording checked and corrected (typo)

- Wording corrected and replaced harmonized

- Sentence revised

- Wording revised

Many thanks again for taking time to review the manuscript!

Best regards,

Michael and Margaux

Kiel/Basel, the 21st of October 2022

##########

For the revision R1, we used the template and hence there is no track-changes version of the MS. We hope, however, that all minor changes are visible in the MS (e.g. the section model set-up (formerly 2.4.2). 

The PLOS ONE system further requested copyrights for the images 1, 2, 3, however the images do not contain particularly protected spatial information and are only reused visualizations. 

Eventually, we confirm the updated funding statements:

MK received funding from the European Commission and the Masaryk University, Brno; Czech Republic, grant number CZ.02.2.69/0.0/0./18_053/0016952; Postdoc2MUNIm order number 21 0053

MK received funding from the German Research Foundation (DFG) and the CRC1266 Scales of Transformation, Kiel University, grant number 290391021

MLCD received funding from the University of Basel and the Swiss National Science Foundation (SNSF).

Many thanks again for considering our work for publication in PLOS ONE and we are looking forward to the publication process!

---

## [Decision Letter · Decision Letter 1]

27 Dec 2022

Scales of transformations - Modelling settlement and land-use dynamics in Late Antique and Early Medieval Basel, Switzerland

PONE-D-22-18464R1

Dear Dr. Kempf,

We’re pleased to inform you that your manuscript has been judged scientifically suitable for publication and will be formally accepted for publication once it meets all outstanding technical requirements.

Kind regards,

Philippe De Smedt

Academic Editor

PLOS ONE

Additional Editor Comments (optional):

Reviewers' comments:

Reviewer's Responses to Questions

**Comments to the Author**

1. If the authors have adequately addressed your comments raised in a previous round of review and you feel that this manuscript is now acceptable for publication, you may indicate that here to bypass the “Comments to the Author” section, enter your conflict of interest statement in the “Confidential to Editor” section, and submit your "Accept" recommendation.

Reviewer #2: All comments have been addressed

2. Is the manuscript technically sound, and do the data support the conclusions?

Reviewer #2: Yes

3. Has the statistical analysis been performed appropriately and rigorously? 

Reviewer #2: Yes

4. Have the authors made all data underlying the findings in their manuscript fully available?

Reviewer #2: Yes

5. Is the manuscript presented in an intelligible fashion and written in standard English?

Reviewer #2: Yes

6. Review Comments to the Author

Reviewer #2: The article now looks good; all suggested changes have been incorporated. I congratulate the authors on this really interesting article.

7. PLOS authors have the option to publish the peer review history of their article (what does this mean?). If published, this will include your full peer review and any attached files.

Reviewer #2: No

---

## [Editor Report · Acceptance letter]

9 Jan 2023

PONE-D-22-18464R1 

Scales of transformations - Modelling settlement and land-use dynamics in Late Antique and Early Medieval Basel, Switzerland 

Dear Dr. Kempf:

I'm pleased to inform you that your manuscript has been deemed suitable for publication in PLOS ONE. Congratulations! Your manuscript is now with our production department. 

Kind regards, 

on behalf of

Dr. Philippe De Smedt 

Academic Editor

PLOS ONE